# Dual fragmentation via collision-induced and oxygen attachment dissociations using water and its radicals for C=C position-resolved lipidomics
Hiroaki Takeda [1,2] ✉, Mami Okamoto[3], Hidenori Takahashi [3], Bujinlkham Buyantogtokh[1], Noriyuki Kishi[2,4], Hideyuki Okano[2,4], Hiroyuki Kamiguchi [2] & Hiroshi Tsugawa [1,5,6] ✉

Oxygen attachment dissociation (OAD) is a tandem mass spectrometry (MS/MS) technique for annotating the positions of double bonds (C=C) in complex lipids. Although OAD has been used for untargeted lipidomics, its availability has been limited to the positive ion mode, requiring the independent use of a collision-induced dissociation (CID) method. In this study, we demonstrated the OAD MS/MS technique in the negative-ion mode for profiling phosphatidylserines, phosphatidylglycerols, phosphatidylinositols, and sulfatides, where the fragmentation mechanism remained consistent with that in the positive ion mode. Furthermore, we proposed optimal conditions for the simultaneous acquisition of CID- and OAD-specific fragment ions, termed OAciD, where oxygen atoms and hydroxy radicals facilitate C=C position-specific fragmentation, while residual water vapor induces cleavage of low-energy covalent bonds as observed in CID. Finally, theoretical fragment ions were implemented in MS-DIAL 5 to accelerate C=C position-resolved untargeted lipidomics. The OAciD methodology was used to illuminate brain region-specific marmoset lipidomes with C=C positional information, including the estimation of C=C positional isomer ratios. We also characterized the profiles of polyunsaturated fatty acid-containing lipids, finding that lipids containing omega-3 fatty acids were enriched in the cerebellum, whereas those containing omega-6 fatty acids were more abundant in the hippocampus and frontal lobe.

Lipids are essential components of cell membranes, signal transduction pathways, and stored energy sources[1–3]. They consist of a backbone (e.g., glycerol, sphingosine, or sterol), a polar head group (e.g., phosphocholine), and fatty acyl chains that vary in length and number of unsaturated bonds. These fatty acyl chains exist in various isomers, differing in double bond (C=C) positions, *sn*-positions, and cis-trans configurations, contributing to the ~50,000 lipids registered in the LIPID MAPS Structure Database[4]. Dysregulation of lipid species abundance disrupts cellular homeostasis and may contribute to disease pathogenesis in animals[5].

Recently, untargeted lipidomics has gained attention for capturing lipid alterations and understanding the potential link between lipid metabolism and phenotypes[6]. Electrospray ionization (ESI) coupled with collision-induced dissociation (CID)-based tandem mass spectrometry (MS/MS) is commonly used to characterize lipid structures. While CID-based fragmentation enables the molecular species-level annotation of lipids, such as determining polar head groups (e.g., phosphocholine of phosphatidylcholine (PC) in the positive ion mode) and total fatty acyl compositions (e.g., fatty acyl chains in the negative ion mode), it does not provide information on C=C positions in fatty acyl chains.

To address this limitation, various analytical methods have been developed to elucidate the detailed structures of fatty acyl chains in complex lipids, a field now referred to as structural lipidomics. Charge-switch derivatization with *N*-(4-aminomethylphenyl) pyridinium not only enhances the sensitivity of fatty acids (FAs) but also provides multiple diagnostic

[1]Department of Biotechnology and Life Science, Tokyo University of Agriculture and Technology, Koganei, Tokyo, Japan. [2]RIKEN Center for Brain Science, Wako, Saitama, Japan. [3]Shimadzu Corporation, 1 Nishinokyo-Kuwabaracho Nakagyo-ku, Kyoto, Japan. [4]Keio Regenerative Medicine Research Center, Kawasaki, Kanagawa, Japan. [5]RIKEN Center for Sustainable Resource Science, Yokohama, Kanagawa, Japan. [6]RIKEN Center for Integrative Medical Sciences, Yokohama, Kanagawa, Japan. ✉e-mail: hiroaki.takeda@aist.go.jp; htsugawa@go.tuat.ac.jp

fragment ions for the annotation of C=C positions[7]. Additionally, the Paternò-Büchi reaction and epoxidation using *meta*-chloroperoxybenzoic acid are effective pretreatment methods for determining C=C positions[8,9]. In gas-phase ion/ion reactions, oppositely charged ions are simultaneously introduced into the ion source of a mass spectrometer, facilitating electron, proton, or chemical group exchange. These reactions generate fragment ions that provide detailed structural information, including FA linkage sites in PCs[10], isomeric PCs and phosphatidylethanolamines (PEs)[11], and C=C positions in FAs[12], offering valuable insights into lipid biochemistry and metabolism. Charge inversion techniques using tris-phenanthroline magnesium dications enable C=C position assignment and quantification of isomeric fatty acyls in glycerophospholipids[13]. Recently, these technologies have been applied to imaging analyses of lipids, including PCs[14,15], FAs[16], and sulfatides (SHexCers)[17], advancing in-depth studies of lipid structures and their functions.

In parallel, advanced fragmentation techniques have emerged as powerful tools for in-depth structural lipidomics without requiring derivatization. Electron-activated dissociation (EAD) is a commercially available technique for the positive ion mode. By employing an electron beam with 10 eV kinetic energy, EAD provides insights into C=C and *sn*-positions, and potentially *E/Z* isomers of phospholipids, sphingomyelin (SM), and triacylglycerol (TG)[18–22]. At higher kinetic energy (≥20 eV), often referred to as electron-induced dissociation, this approach enables the characterization of *sn*- and C=C positions in PCs[23] and has been applied to imaging analyses of brain tissue[24–26]. However, the complexity of MS/MS spectra caused by bond cleavage at multiple sites, as observed in EAD, necessitates complete chromatographic separation of isomers to prevent fragment overlap from co-eluting species. EAD is effective for lipids present at concentrations of 1 μg mL$^{-1}$ or higher and relies on the presence of metal ions, such as sodium, for the structural elucidation of negatively charged glycerophospholipids like phosphatidylinositol (PI), phosphatidylglycerol (PG), and phosphatidylserine (PS).

Ozone-induced dissociation (OzID) utilizes gas-phase ion-molecule reactions with ozone vapor to produce product ions specific to double bonds in unsaturated lipid ions[27]. The combined technique of CID and OzID (COzID) has been developed to differentiate *sn*-isomers of alkali-adducted phospholipids by inducing ozonolysis at the newly formed double bonds at the *sn*-2 position, following the neutral loss of the polar head group via CID[28]. OzID has been successfully applied to FAs[29] and phospholipids[30], revealing comprehensive structural insights. Similarly, 193 nm ultraviolet photodissociation (UVPD) determines C=C positions in PCs through photoinduced cleavage of carbon-carbon bonds, producing a characteristic mass difference of 24 Da at the double bond[31]. The hybrid MS$^3$ strategy, which integrates collisional activation and UVPD of polar head loss ions, enables precise localization of fatty acyl chains and their double bonds[32]. Additionally, parallel reaction monitoring via UVPD facilitates the quantification of PC isomers with distinct C=C positions[33].

Recently, radical-induced dissociation methods such as hydrogen abstraction dissociation (HAD) and oxygen attachment dissociation (OAD) have been developed to facilitate structural elucidation. HAD utilizes hydrogen radicals (H·) generated through the thermal dissociation of hydrogen molecules passing through a heated tungsten capillary at temperatures exceeding 2000 °C in the ion trap[34]. Hydrogen abstraction by H· generates hydrogen-deficient radical sites at carbon-carbon bonds in phospholipid acyl chains, which subsequently induce site-specific cleavage. Similar to EAD, HAD generates comprehensive spectra that provide information on both the positions of OH modifications within the carbon chain and the C=C positions[35]. OAD has been developed as a complementary technique that specifically targets C=C positions in molecules of interest, further refining lipid structural analysis. OAD, now commercially available, utilizes atomic oxygen (O) and/or hydroxyl radicals (OH·) generated by the microwave irradiation of water vapor (H$_2$O)[36,37]. These reactive species interact with π-electrons in double bonds (R1-C=C-R2), inducing the cleavages between R1 and C, and C and R2 whose fragment ions can be

used as the diagnostic marker for the C=C position determination in complex lipids. OAD's fragmentation patterns are similar to those of OzID, acquiring C=C position-specific fragment ions, but OAD is easier to handle and safer than ozone, a strong oxidant. In our previous study, we demonstrated C=C position-resolved lipidomics in positive ion mode by integrating molecular species-level annotation obtained from CID MS/MS techniques in both positive and negative ion modes[38]. For example, "PC 16:0_18:1" was first annotated by CID MS/MS, and the C=C position was subsequently determined using OAD MS/MS, resulting in the final annotation "PC 16:0_18:1(Δ9)". While the successful annotation of cationic lipids, such as PC and PE, was shown, the OAD technique is versatile and applicable in both positive and negative ion modes. Although OAD, unlike EAD, cannot determine *sn*-positions, CID-specific fragment ions related to polar head groups and FA compositions can still be obtained under OAD collision-cell conditions.

In this study, we investigated OAD mass fragmentation in both positive and negative ion modes to reveal C=C structural isomers, including negatively charged lipid molecules. We also explored a collision cell condition that generated both CID- and OAD-specific fragment ions using H$_2$O and its radicals, termed OAciD, which enabled in-depth structural elucidation of lipids with a single analytical method. Finally, we developed and implemented an algorithm for automatic lipid annotation based on the quality of the acquired product ion spectra in MS-DIAL 5 software[39,40]. The accuracy of the automated annotation was evaluated using an UltimateSPLASH mixture containing 69 synthetic lipid standards. Using this advanced technique, we performed untargeted lipidomics on the marmoset brain, characterizing the unique lipidomes of the frontal lobe, hippocampus, midbrain, cerebellum, and medulla.

## Results
### Extension of lipid coverage by using both positive and negative ion modes

To extend lipid coverage, we evaluated the fragmentation patterns and efficiency of OAD MS/MS in negative ion mode, marking its first application in OAD-based lipidomics for this polarity. Additionally, we examined sodium adduct ions, as recommended in a previous lipidomics study under EAD conditions[22], to efficiently determine the C=C positions of negatively charged lipids such as PG and PI. The use of organic solvents containing sodium acetate, introduced post-column, increases the formation and sensitivity of sodium adduct ions[41]. However, employing sodium salts presents several challenges, including increased background noise in the total ion chromatogram, reduced sensitivity and reproducibility of conventional proton and ammonium adduct ions, and mass spectrometer contamination due to sodium's nonvolatility.

We compared the sensitivity of the MS$^1$ peaks under three ionization conditions: conventional proton or ammonium adduct ions in the positive ion mode ([M + H]$^+$ or [M + NH$_4$]$^+$), deprotonated or acetate adduct ions in the negative ion mode ([M − H]$^-$ or [M + CH$_3$COO]$^-$), and sodium adduct ions in the positive ion mode ([M+Na]$^+$). The column eluent (600 μL min$^{-1}$ total) was mixed with 50 μL min$^{-1}$ of 50% methanol containing 200 μM sodium acetate post-column to form the sodium adduct ion (Fig. S1a). Although the addition of sodium acetate solution increased background noise (Fig. S1b), the peak area reproducibility of proton, ammonium, and sodium adduct ions was within 10% for each lipid subclass when 50 μL min$^{-1}$ of sodium solution was mixed post-column (Fig. S1c). Among lipid subclasses, diacylglycerol (DG) and ceramide (Cer) showed improved sensitivities with the sodium adduct form (Fig. S1d). In contrast, the negative ion mode provided higher sensitivity for detecting PG and PI (Fig. S1d).

We further evaluated the sensitivity of the MS/MS peaks using a product ion scan in which appropriate precursor ions were selected based on the full MS scan data (Supplementary Data 1). In the fragmentation of FA 18:1(Δ9) fatty acyl moieties, neutral losses of 96.13 Da and 138.14 Da, corresponding to the cleavages of Δ10-Δ11 and Δ8-Δ9 positions,

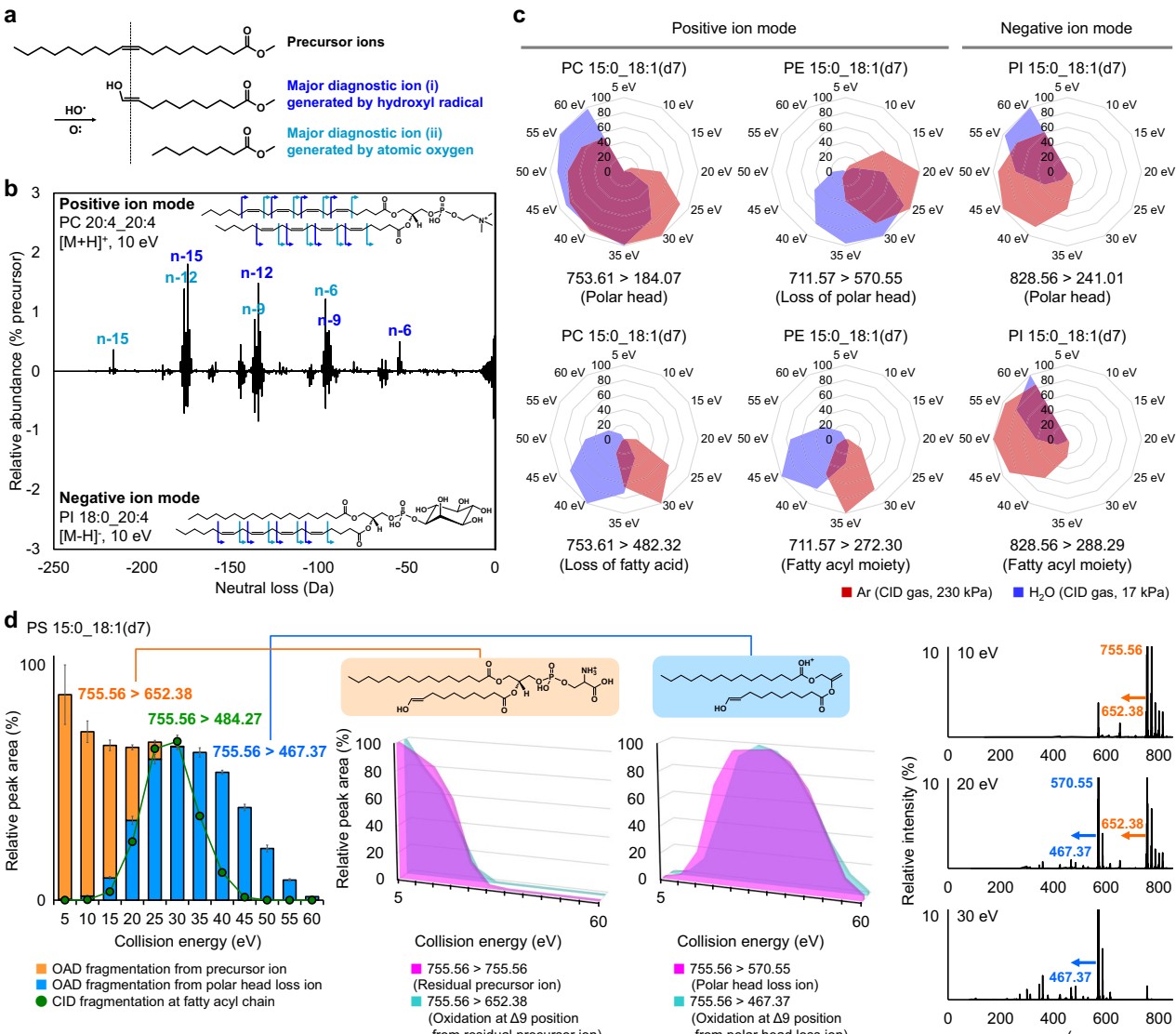

**Fig. 1 | Characteristics of OAciD MS/MS. a** Major fragmentation of OAD MS/MS. **b** Fragmentation difference between positive and negative ion modes. **c** Appropriate collision energy in Ar gas and $H_2O$ vapor. The area value of $MS^2$ peak was normalized to the maximum peak detected under the collision energy range of 5–60 eV. All lipid subclasses are summarized in Fig. S2. **d** Fragmentation at the C=C position by increasing the collision energy. In the left panel, the orange and blue bars represent the $MS^2$ peak areas of fragments at the C=C position, derived from the residual precursor and polar head loss ions, respectively. The ion at $m/z$ 467.37 originated from the neutral loss of the polar head group ($m/z$ 570.55); however, the notation "755.56 > 467.37" is used, as $m/z$ 467.37 is a product ion generated from the precursor $m/z$ 755.56. The green line plot indicates fragments from fatty acyl chains, also observed in CID mode using Ar gas. Error bars represent the standard deviations of four analytical replicates. Relative peak areas (%) denote the ion intensities in the product ion spectrum, with the most intense peak set to 100%. A summary of all lipid subclasses is provided in Fig. S3. The middle panels show the $MS^2$ peak areas of fragments at the C=C position from residual precursor and polar head loss ions. A summary of all lipid subclasses is provided in Fig. S4. The right panels display the MS/MS spectra at 10, 20, and 30 eV.

respectively, were observed (Fig. 1a)[38]. No differences in fragmentation patterns were detected when comparing MS/MS spectra between positive and negative ion modes, indicating that the fragmentation mechanism was unaffected by polarity differences (Fig. 1b). However, the ratio of fragmentation efficiency between positive and negative ion modes varied across lipid subclasses. The highest MS/MS peaks for lyso-PC (LPC), lyso-PE (LPE), PC, SM, DG, and TG were observed in sodium adduct forms, whereas better sensitivities were achieved in the negative ion mode for PG, PI, and Cer (Fig. S1d). The high fragmentation efficiency of LPC, LPE, PC, SM, DG, and TG in the sodium adduct form is likely due to the strong charge localization introduced by metal ions like sodium, compared to proton or ammonium ions.

Nevertheless, we chose to use protons and ammonium ions because data-dependent MS/MS acquisition in untargeted lipidomics is more efficient when the signal-to-noise ratio is higher in full scan $MS^1$. Based on our results, an ammonium acetate-containing LC solvent, which efficiently detects protons or ammonium ions, facilitates data-dependent MS/MS-based lipidomics for screening purposes. Meanwhile, using sodium adduct forms in targeted MS/MS mode would accelerate in-depth structural elucidation. In this study, as our aim was to develop a lipidomics pipeline using data-dependent MS/MS via OAciD, we optimized the analytical method using proton and ammonium ions, enabling lipid quantification through $MS^1$ peaks and annotation via data-dependent MS/MS.

## Dual fragmentation of CID and OAD using $H_2O$ and its radicals

Fragment ions of polar head groups and fatty acyl moieties are essential for annotating lipid molecules using MS/MS. For instance, the phosphorylcholine fragment ion ($m/z$ 184.07) detected in the positive ion mode is useful for annotating LPC, PC, and SM. However, the MS/MS intensity of these polar head-specific fragment ions was insufficient in previous OAD

MS/MS systems because the collision cell utilized only 5–10 eV, which is adequate for ion transfer but not for efficient fragmentation. We hypothesized that by increasing the collision energy within the collision cell, fragment ions from simultaneous OAD and CID reactions could be observed, as typical lipid product ion spectra contain a portion of unreacted ions (i.e., precursor ions).

Inert gases, such as argon (Ar) and nitrogen, are typically used in CID MS/MS to minimize unwanted reactions in collision cells. However, OAD MS/MS employs O and/or OH radicals generated by microwave irradiation of $H_2O$ vapor for radical reactions. Therefore, it is necessary to obtain fragment ions of polar head groups and fatty acyl moieties using residual $H_2O$ vapor instead of inert gases in this system. We first investigated the differences in fragmentation patterns between $H_2O$ vapor and Ar (Supplementary Data 2). The results showed that the appropriate collision energy for generating product ions related to polar head groups and FA chains was on average 10 eV higher with $H_2O$ vapor than with Ar gas (Figs. 1c and S2), although the fragmentation patterns remained similar under both conditions.

We further explored the sensitivity of fragment ions derived from the C=C position (i.e., OAD reactions) by varying the collision energy from 5 to 60 eV. The product ion of the C=C position-related fragment of PC decreased at collision energies above 20 eV, while both fatty acyl-related fragment ions ($m/z$ 482.32) and C=C position-related fragment ions ($m/z$ 650.44) were observed at 30–35 eV (Fig. S3a). Notably, when comparing the $MS^2$ peak areas of the residual precursor and the C=C position-related fragment ion, the sensitivities were highly correlated with collision energy conditions (Fig. S4a), indicating that the sensitivity of C=C position-related fragment ions depends on the ion abundance of the residual precursor. Neutral-loss ions of polar head groups, such as those observed in PE, PG, and PS, were detected with increased collision energy (Fig. S2). Two types of C=C position-related fragment ions were observed in these subclasses: one derived from the precursor ion and the other from the product ion generated by the neutral loss of the polar head groups (Figs. 1d and S3a). The total peak areas of these oxidative fragments from the residual precursor and neutral loss ions were nearly identical (Figs. 1d and S3a). Since the OAD reaction is slower than the CID reaction, OAciD MS/MS with higher collision energy produces spectra that can be obtained in $MS^3$, enabling sequential fragmentation from the polar-head-related neutral loss ion to the C=C position-specific ion. The strong correlation between the neutral-loss fragments of polar head groups and the oxidative fragments at the C=C position from these neutral-loss fragments further confirms that the OAD reaction does not depend on collision energy (Figs. 1d and S4).

Additionally, we found that a collision energy of 30 eV was optimal for generating both C=C position (OAD) and fatty acyl composition (CID)-related fragment ions (Figs. 1d and S2–S4). Thus, we used 30 eV as the collision energy for the dual fragmentation of OAD and CID (OAciD) in positive ion mode. The collision energy in negative ion mode should be optimized to efficiently detect the PI and PG molecules, as their detection was less effective in the positive ion mode. While higher collision energy is required when annotating the fatty acyl composition of PI (Fig. 1c), we also set the collision energy at 30 eV in the negative ion mode analysis, balancing the sensitivity for C=C position (OAD) and fatty acyl composition (CID) in PI molecules ($m/z$ 241.01, 0.61 ± 0.13%; $m/z$ 288.29, 0.15 ± 0.03%; $m/z$ 683.37, 0.18 ± 0.05%; and $m/z$ 725.39, 0.37 ± 0.08% from the residual precursor intensity) (Figs. S2b and S3b). The MS/MS spectra of PC and PI obtained using this optimized method are shown in Fig. 2a. Despite the complexity introduced by $H_2O$ vapor adduct ions, our findings indicate that using 30 eV in the collision cell enables the simultaneous detection of polar head, FAs, and C=C position-specific fragment ions in a single analysis.

## Evaluation of OAciD MS/MS spectral annotations using MS-DIAL 5

The dynamic range of lipid quantification was investigated using the EquiSPLASH standard containing 100 µg mL$^{-1}$ of 13 deuterium-labeled synthetic lipids with a double bond at the Δ9 position except for Cer

18:1;O2/15:0 (Supplementary Data 3). Owing to the narrow linear range of the fragment ions (less than two digits), the lipids were quantified using $MS^1$ peaks, and their detailed structures were characterized using fragments obtained by data-dependent MS/MS. To increase the number of $MS^1$ peaks that can characterize the C=C position while maintaining the linear range of the $MS^1$ peaks, $MS^1$ and $MS^2$ ion accumulations were turned off and on, respectively, in the Shimadzu MS setting. The C=C position could also be characterized at low concentrations in the linear range using the positive ion mode (≥50 ng mL$^{-1}$ for PC) (Fig. 2b). However, the sensitivity of C=C position-related fragment ions in the OAciD MS/MS spectra in negative ion mode is still challenging, although OAciD MS/MS works for the PI and PG targeted in this study (Fig. 2b). In addition, the MS-DIAL 5 software program was updated for fragment annotation of the OAciD MS/MS spectra. The program uses a rule-based (decision-tree-based) algorithm, in which the existence of diagnostic ions related to the polar head, fatty acyls, and C=C positions are confirmed. Importantly, MS-DIAL requires molecular species-level (MSL) annotations such as PC 16:0_18:1 to determine the C=C position. Furthermore, the program excluded candidates if the major diagnostic fragment ions related to the C=C position were not detected in the MS/MS spectrum. For example, neutral losses of 96.13 Da (Δ10-Δ11 cleavage) and 138.14 Da (Δ8-Δ9 cleavage) were used for the diagnostic fragments of PC 16:0_18:1(Δ9). Finally, candidates with all diagnostic fragment ions in the MS/MS spectrum were ranked based on the correlation coefficient between the experimental and computationally generated in silico MS/MS spectra.

We evaluated the annotation accuracy of MS-DIAL 5 using liquid chromatography-mass spectrometry (LC-MS) data obtained from a dilution series of UltimateSPLASH containing 69 deuterium-labeled synthetic lipids at various concentrations. This standard mixture contained varying numbers of double bonds in the acyl chains of FAs, including polyunsaturated FA (PUFA). Under the Shimadzu LC-MS/MS conditions employed in this study, significant in-source fragmentation prevented the detection of cholesteryl esters (CEs). Although lowering the interface temperature (<150 °C) could potentially mitigate this issue, it would simultaneously compromise the sensitivity of other lipid subclasses. To maintain analytical robustness, CE data were therefore excluded from the evaluation. Among the nine lipid subclasses, the fatty acyl compositions and their C=C positions were accurately determined in positive ion mode, except for DG, SM, and some PE and TG molecules (Figs. 2c and S5a). Of these, the annotation results of SM and PE molecules became species level (SL) because the $O$-acyl and $N$-acyl fragment ions used for MSL annotations were difficult to detect in the MS/MS spectra, whereas the C=C-specific fragment ions were clearly detected. The sensitivity of the fragment ion ($m/z$ 264.27) for $N$-acyl chain determination of the SM subclass was particularly low at 30 eV (Fig. S2a). Thus, we further evaluated the MS-DIAL 5 program in cases where MSL annotations for all lipid subclasses were pre-defined before the C=C position determination to examine only the OAD spectrum (Figs. 2d and S5b). For example, when annotating the spectrum of PC_d5 17:0_16:1(Δ9), the peak was manually assigned to PC_d5 17:0_16:1 in MS-DIAL, ensuring that MSL annotation was established before initiating C=C position searches. This approach is executable in the current version of MS-DIAL through a user-defined text library containing compound names, precursor $m/z$ values, and retention time information. The results demonstrated that MS-DIAL 5 accurately annotated the C=C positions of phospholipids and SM when the acyl chain information was provided. The results of misannotations in several TG molecules indicated that the interpretation of OAciD MS/MS spectra for the ammonium adduct form of TG remains difficult when various unsaturated FAs are contained in the TG molecules. Because the optimal collision energy conditions for obtaining MSL annotations differ widely among lipid subclasses, the collision energy conditions should be changed for profiling the targeted lipid subclasses. Nevertheless, the MS-DIAL 5 program automatically annotates OAciD MS/MS spectral data, in which an appropriate lipid description is generated based on the quality of the product ion spectrum. The same was true in the

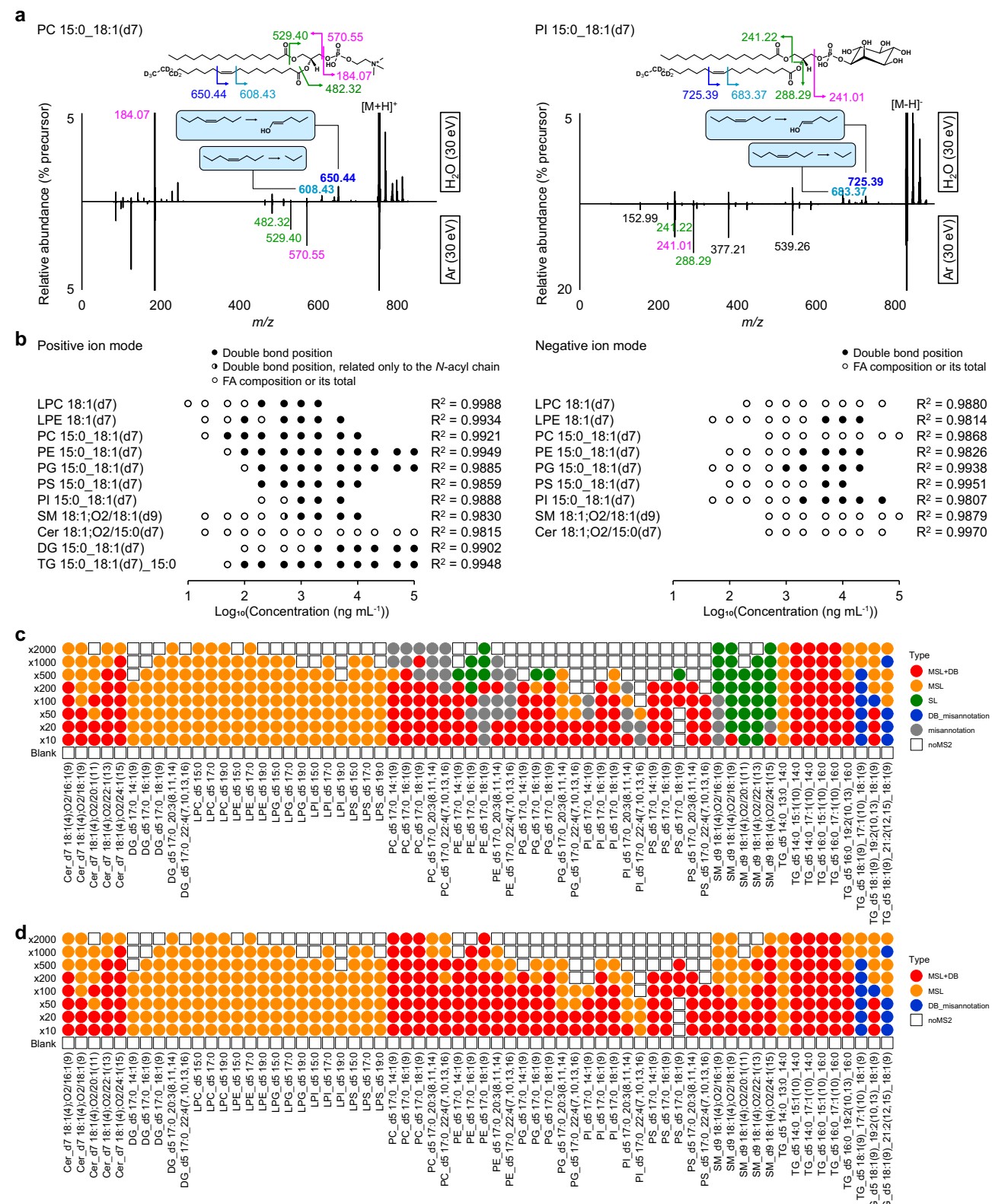

evaluation for negative ion mode spectra, where the anionic lipids, especially PI molecules were successfully characterized (Fig. S5c, d). On the other hand, the product ion spectra related to C=C positions of PC and Cer molecules were not observed clearly (Fig. 2b), suggesting that these molecules should be characterized in positive ion mode.

## C=C position-resolved in-depth lipidomics of marmoset brain

Marmosets have attracted considerable attention in neuroscience research because of their neurological and genetic similarities to humans. Untargeted lipidomics has attracted attention for understanding the complexity of brain lipids and their changes during aging[42–44]; however, little work has been done

**Fig. 2 | Automated annotation of lipid standards using MS-DIAL 5 software.**
**a** Summary of MS/MS spectra by setting 30 eV of collision energy in both positive and negative ion modes. **b** Calibration curve of each lipid subclass in positive and negative ion modes (n = 4). Data-dependent MS/MS mode was used for evaluation. Circles represent the linear range of the MS[1] peak with R[2] > 0.9800 (Supplementary Data 3). The fully filled circles represent concentrations where the C=C position could be determined, with two major fragments detected simultaneously in at least two out of four analyses. The half-filled circle for SM indicates that the C=C position was determined only on the *N*-acyl chain, not the long chain base. Annotation depth was assessed using the EquiSPLASH mixture containing deuterium-labeled standards. **c** Automated annotation results of 69 synthetic lipid standards using the optimized MS-DIAL 5 software. The UltimateSPLASH standard mixture was diluted 10, 20, 50, 100, 200, 500, 1000, and 2000 times with MeOH. For example, "x10" indicates a dilution 10 times less concentrated than the original, denoted as "x1". Colors represent the structural depth of the automatic annotation: red indicates both fatty acyl composition and C=C position (MSL + DB); orange indicates fatty acyl composition only (MSL); green indicates species level annotation only (SL); blue

indicates the misannotation of C=C position (DB_misannotation). If the MS/MS spectrum was not assigned to the precursor ion by data-dependent acquisition, a square shape with a black color is used (noMS[2]) The representative annotation was determined as follows: if the same lipid name was annotated in at least two of the four replicates, that name was used as the representative annotation. If the annotation results differed across all four replicates, the lipid with the highest score was adopted as the representative. In this study, fragment ions corresponding to the sphingobase Δ4 C=C positions were not detected under the applied collision energy settings. However, C=C position-specific fragment ions from *N*-acyl chains were detected, as confirmed using UltimateSPLASH standards. Therefore, the label "MSL + DB" was assigned to sphingolipids when the *N*-acyl chain C=C positions were determined. Lysophospholipids such as LPC, LPE, LPG, lyso-PI (LPI), and lyso-PS (LPS) are out of range for C=C position determination due to their saturated fatty acyl moieties. **d** Annotation results using the MS-DIAL 5 program when the MSL annotations were carried out before the C=C positional annotations were performed. The definitions of color and symbol are the same as used in Fig. 2c.

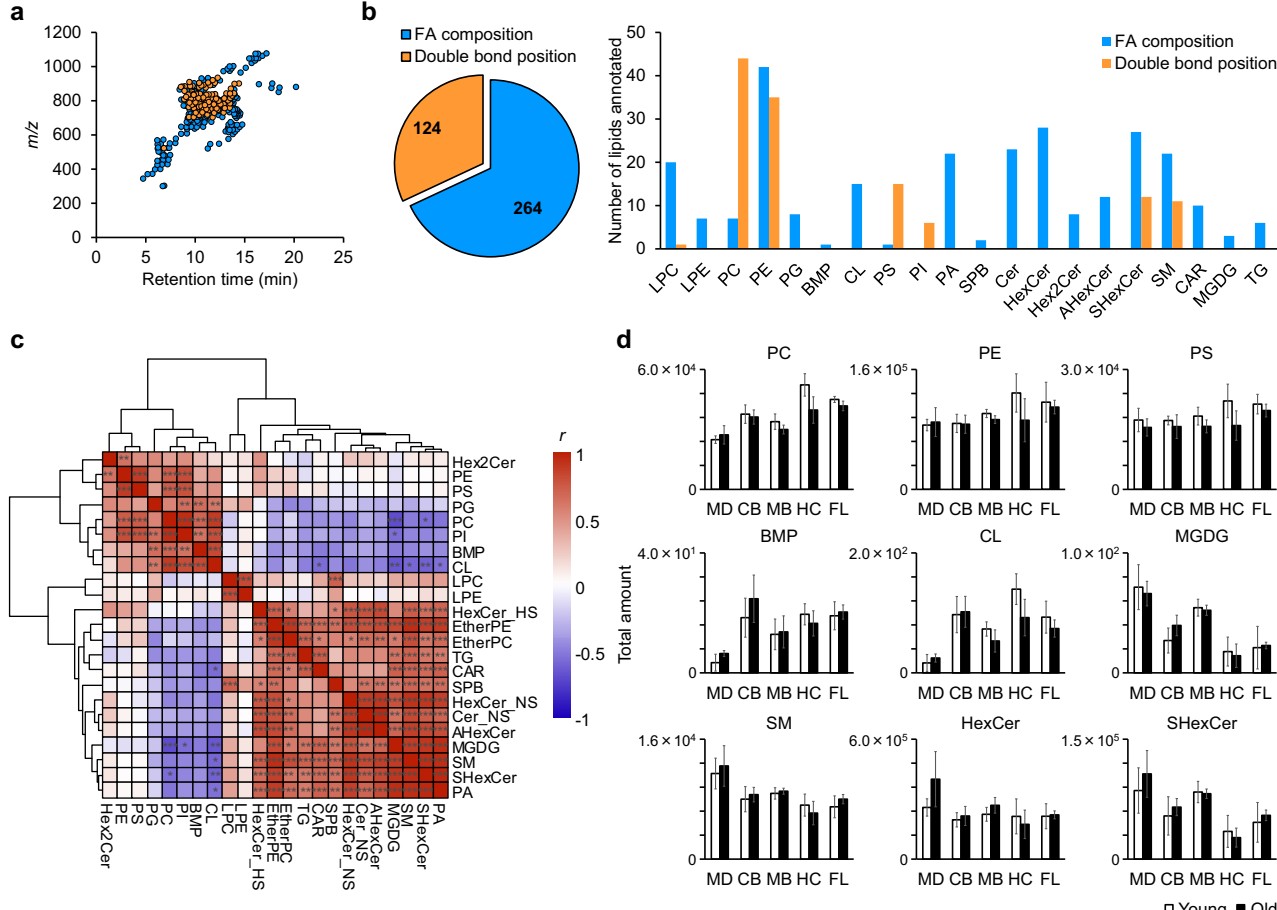

**Fig. 3 | Untargeted lipidomics of marmoset brain sections. a** Scatter plot of lipid molecules detected in marmoset brain section. **b** Number of lipids detected at the structural depth of fatty acyl compositions or C=C positions. **c** Hierarchical clustering analysis using the correlation coefficient between lipid subclasses. A total of 40 samples were used for the correlation analysis without discriminating between different regions and ages to characterize lipid subclasses. Correlation analyses were performed using Pearson correlation coefficient. The *p* values obtained from these

analyses were adjusted for multiple comparisons using the false discovery rate method. Adjusted *p*-values were classified into significance levels and displayed using symbols: *** for $p < 0.001$, ** for $p < 0.01$, and * for $p < 0.05$. **d** Total amounts of representative lipid subclasses in each region and age. Five regions including frontal lobe (FL), hippocampus (HC), midbrain (MB), cerebellum (CB), and medulla (MD) were analyzed. Error bars indicate the standard deviations of the four biological replicates in each group.

on reversed-phase LC (RPLC)/MS-based untargeted lipidomics of marmoset brain regions. We applied our structural lipidomics approach to marmoset brain sections using RPLC coupled to OAciD MS/MS system, including the frontal lobe, hippocampus, midbrain, cerebellum, and medulla (Supplementary Data 4). The total ion chromatograms for each marmoset brain region are shown in Fig. S6a. In total, 388 lipids were

detected in the marmoset brain, of which 124 molecules were annotated with C=C positions (Fig. 3a). The fragmentation behaviors of OAciD MS/MS, that is, the balance of FA compositions and their C=C positions, were also demonstrated in biological samples, in which most of the PC, PS, and PI molecules were annotated at the structural depth of the C=C positions (Fig. 3b). Additionally, the C=C positions of SHexCers on the sphingosine

backbone were annotated based on the fragmentation rules of Cers optimized using standard mixtures of both EquiSPLASH and UltimateSPLASH. The versatility of this fragmentation method was demonstrated by annotating 20 lipid subclasses, including minor lipids, such as sphingoid base (SPB), acylhexosylceramide (AHexCer), acylcarnitine (CAR), and monogalactosyl DG (MGDG) (Fig. 3b and Supplementary Data 4).

The localization of lipid subclasses in the marmoset brain is similar to that in the mouse brain[42,43]. Gene expression in the marmoset brain involved in the lipid subclasses was searched using the Marmoset Gene Atlas[45,46] (Fig. S6b). Hierarchical clustering analysis using the correlation coefficients between lipid subclasses identified two groups consisting mainly of phospholipids and sphingolipids (Fig. 3c). Hexosylceramides (HexCers) and SHexCers are enriched in oligodendrocytes that form myelin sheaths on the axons of neurons[42,43]. These lipid subclasses were slightly higher in the medulla than in other regions (Fig. 3d). In addition to these sphingolipids, AHexCers and MGDGs clustered in the same group. MGDGs and galactosylceramides are synthesized from DGs and Cers through catalysis by UDP-galactose-ceramide galactosyltransferase (CGT)[47]. CGT is expressed in oligodendrocytes and Schwann cells, which is consistent with the high localization of galactosylceramides. In situ hybridization of the myelin basic protein (MBP) gene, one of the major components of oligodendrocytes, showed high expression in the midbrain and medulla and low expression in the hippocampus and cerebellum (Fig. S6b). The localization of glycolipids, such as SHexCer and MGDG, was similar to that of *Mbp*. Phospholipids, except lysophospholipids, ether PC, ether PE, and phosphatidic acid (PA), clustered in the same group (Fig. 3c). Among them, major phospholipids, such as PC, PE, PS, and PI, were correlated, but their specific localization was not observed, suggesting that the phospholipid components of cell membranes were almost constant despite the presence of different cell types. In contrast, low levels of bis(monoacylglycero)phosphate (BMP) and cardiolipin (CL) were observed in the medulla (Fig. 3d). BMP, which is synthesized from a lyso-PG (LPG) substrate by ceroid-lipofuscinosis neuronal 5 (CLN-5), is localized in organelles such as late endosomes and lysosomes[48]. Consistent with this localization, *Cd63*, a late endosome marker, was expressed at low levels in the medulla, as determined by in situ hybridization of the marmoset brain (Fig. S6b). CL is a specific lipid subclass in the inner mitochondrial membrane that plays structural and functional roles such as maintaining mitochondrial morphology and regulating apoptosis[49]. A correlation between CL and the mitochondrial integral membrane protein TOMM20 was observed in the rat brain[49]. High expression of *Tomm20* in the cornu ammonis and dentate gyrus of the hippocampus was observed by in situ hybridization (Fig. S6b), and high levels of total CLs were detected in the hippocampal region using our lipidomics data (Fig. 3d). Consequently, our results indicated that OAciD MS/MS represented reasonable localization of lipid subclasses reported in previous brain lipidomics study of mouse[42,43] and human[44].

Several structural isomers, including those at different C=C positions, have been observed in the marmoset brain. Although a single MS[1] peak of PC 16:0_16:1 was observed, MS/MS fragments of the C=C position suggested the co-elution of PC 16:0_16:1($\Delta$7) and PC 16:0_16:1($\Delta$9) (Fig. 4a). OAciD MS/MS offers a distinct advantage over EAD when analyzing co-eluted C=C isomers. For instance, in the case of PC 16:0_16:1($\Delta$7) and PC 16:0_16:1($\Delta$9), EAD MS/MS annotation requires both diagnostic *m/z* values and the characteristic V-shaped pattern of product ion intensities. This pattern results from highly efficient fragment formation via homolytic cleavage at the allylic position (stabilized by delocalized electrons) and the McLafferty rearrangement (involving hydrogen transfer at the $\gamma$-position). Since electrons disrupt each carbon-carbon covalent bond in fatty acyl chains, producing H-loss, radical, and H-gain fragment ions, the characteristic V-shaped pattern is often obscured when isomers co-elute (Fig. S6c). In contrast, OAciD produces distinct, non-overlapping diagnostic fragment ions, enabling more precise annotation of these isomers (Figs. 4a and S7). This advantage is further highlighted in abundance ratio estimations of co-eluted isomers, as standard deviations in diagnostic product ion intensity ratios are larger in EAD MS/MS than in OAD MS/MS[40].

Moreover, some MS[1] peaks of PC 18:0_22:5 were detected, of which these MS/MS spectra represented PC 18:0_22:5($\Delta$4,7,10,13,16) and PC 18:0_22:5($\Delta$7,10,13,16,19), respectively (Fig. 4a). Hierarchical clustering analysis of the constituent PC molecules clearly classified structure-dependent groups, such as chain length, odd or even chains, saturated or unsaturated bonds, and C=C positions (Fig. 4b). For *n*-9 monounsaturated FAs (MUFAs), the trend differed for chain lengths greater than or less than 20 carbon atoms. PC containing long MUFAs such as PC 18:1($\Delta$9)_24:1($\Delta$15) were localized in the medulla, cerebellum, and midbrain (Fig. S6d). Among the *n*-6 PUFAs, PCs containing arachidonic acid (20:4, *n*-6), docosatetraenoic acid (22:4, *n*-6), and docosapentaenoic acid (22:5, *n*-6) were clustered in the same group; however, they did not correlate with those containing linoleic acid (18:2, *n*-6), even though the constituent FAs were placed in the same FA metabolic pathway. Different patterns of fatty acyl composition and specific localization of some constituent molecules have been determined in the marmoset brain. Although eicosapentaenoic acid (EPA) (20:5, *n*-3) and docosahexaenoic acid (DHA) (22:6, *n*-3) are in the same synthetic pathway as *n*-3 PUFAs, EPAs levels are very low compared to DHAs in the brain. Only PC 16:0_20:5($\Delta$5,8,11,14,17) was detected as an EPA-containing phospholipid also in the marmoset brain (Supplementary Data 4). The localization of phospholipids was mainly dependent on the fatty acyl compositions rather than on the lipid subclasses. High levels of saturated PC and PE were detected in the frontal lobe, and low levels were detected in the medulla (Fig. S6d). The most specific localization was observed for phospholipids containing diacyl PUFAs. Phospholipids containing diacyl *n*-3 PUFAs are localized in the cerebellum, where the amounts of PS 22:5($\Delta$7,10,13,16,19)_22:6($\Delta$4,7,10,13,16,19) were especially enriched when compared with the other tissues (Figs. 4c and S6e).

Finally, we characterized the ratios of PC 16:0_16:1($\Delta$7) to PC 16:0_16:1($\Delta$9) and PC 16:0_18:1($\Delta$9) to PC 16:0_18:1($\Delta$11) using spectral data of data-dependent MS/MS via OAciD (Fig. 4d). According to our previous study[40], the C=C position-specific fragmentation efficiency between PC 16:0_18:1($\Delta$9) and PC 16:0_18:1($\Delta$11) was nearly equal, allowing the relative abundance of these lipid isomers to be estimated by MS/MS peak heights. Therefore, the product ion peak heights of *m/z* 636.42 and *m/z* 664.45, which were not interfered with co-eluting isomers, were used to calculate the ratio of PC 16:0_16:1($\Delta$7) to PC 16:0_16:1($\Delta$9) (Fig. S7). However, these ratios should be considered semi-quantitative due to the absence of authentic standards for these isomers. The peaks of *m/z* 664.45 and *m/z* 692.49 were used for the ratio of PC 16:0_18:1($\Delta$9) to PC 16:0_18:1($\Delta$11). The results proposed PC 16:0_16:1($\Delta$9) and PC 16:0_18:1($\Delta$9) as the major isomers. FA 16:1($\Delta$9) and FA 18:1($\Delta$9) are biosynthesized by stearoyl coenzyme A desaturase 1 (SCD1) using palmitic acid (FA 16:0) and stearic acid (FA 18:0) as substrates, respectively. FA 18:1($\Delta$11), known as cis-vaccenic acid, is biosynthesized by ELOVL fatty acid elongase 6 (ELOVL6) using FA 16:1($\Delta$9) as a substrate. The FA 16:1($\Delta$7) isomer, rarely reported in mammalian cell studies, may be synthesized through the beta-oxidation of FA 18:1($\Delta$9). Given that the brain is a mitochondria-rich organ, beta-oxidation activity is likely higher compared to other organs, though the mechanistic role of PC 16:0_16:1($\Delta$7) in the brain remains unknown.

## Discussion

In this study, we developed an advanced structural lipidomics method using OAciD MS/MS to determine the C=C position in the fatty acyl chains of diverse lipid subclasses. Our OAciD MS/MS-based approach overcame technical limitations related to lipid subclass coverage and analytical throughput by enabling analysis in negative ion mode and simultaneous dual fragmentation using CID and OAD. Notably, this method eliminates the need for data acquisition of conventional CID lipidomics by achieving simultaneous cleavage via residual $H_2O$ vapor and radical species (O and/or OH·). The OAD MS/MS system offers two key advantages over previous methods for determining C=C positions. First, the OAD reaction is independent of ion mode polarity (positive or negative), although sensitivity varies depending on lipid subclasses and adduct ions. For example, sodium

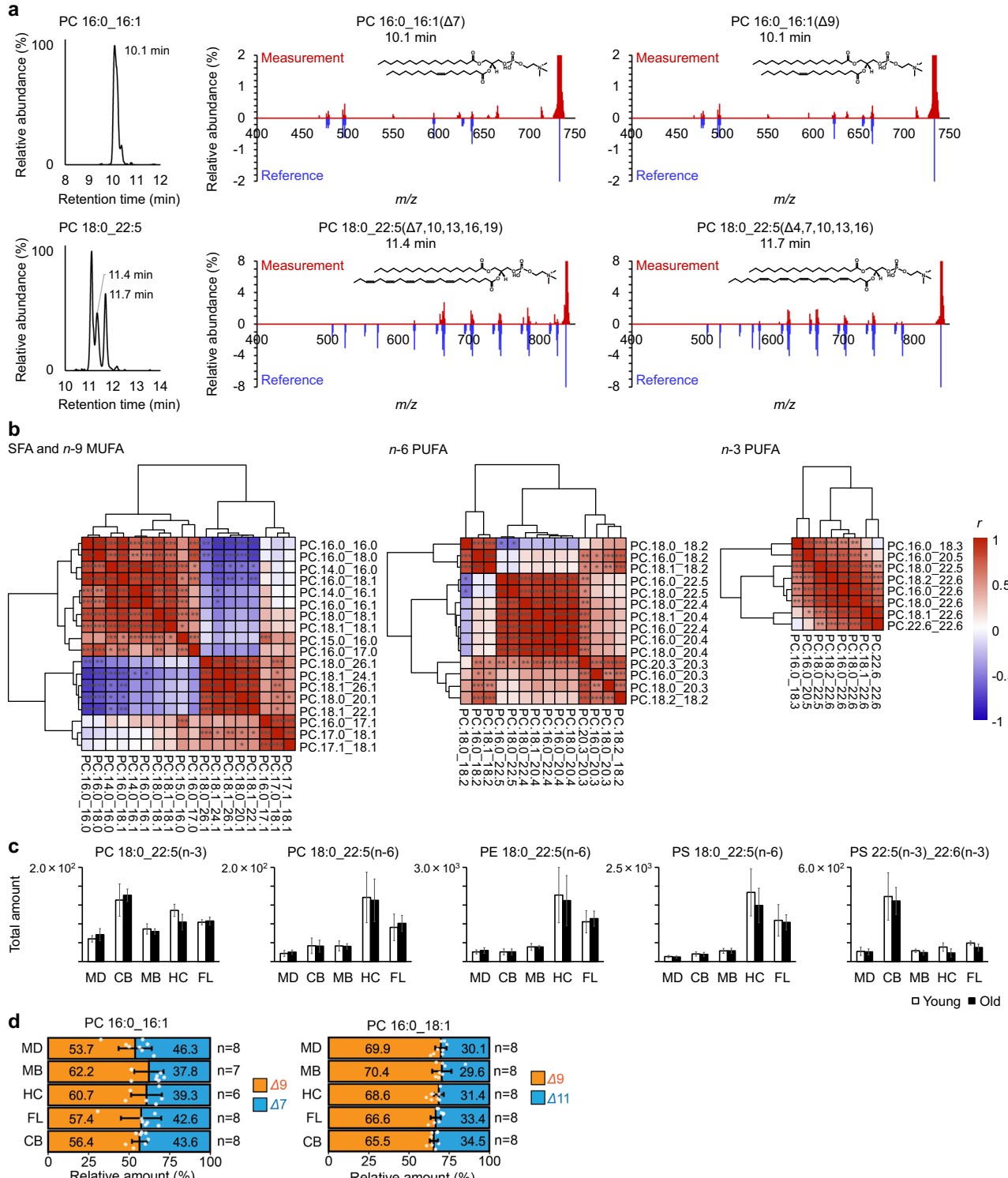

**Fig. 4 | Characterization of lipid molecules at the structural depth of the C=C position. a** Chromatograms and MS/MS spectra of structural isomers with different C=C positions. **b** Hierarchical clustering analysis using the correlation coefficient between PC molecules. Saturated FA (SFA) and *n*-9 MUFA, *n*-6 PUFA, and *n*-3 PUFA were analyzed using the same methods as in Fig. 3c. **c** Total amounts of constituent molecules in each region and age. Error bars indicate the standard deviations of the four biological replicates in each group. **d** The ratios of PC 16:0_16:1(Δ7) and PC 16:0_16:1(Δ9) and ratio of PC 16:0_18:1(Δ9) and PC

16:0_18:1(Δ11) among tissues. Error bars indicate the standard deviations of the biological replicates in each group where both young- and aged marmosets were included. Due to the limit of detection in the data-dependent acquisition mode for PC 16:0_16:1, total three samples containing one midbrain and two medulla samples were excluded. PC 14:0_18:1 isomers may be present as minor contaminants within the peak at 10.1 min; however, their contribution is negligible in marmoset brain regions. Further details are shown in Fig. S7.

ions exhibit strong charge localization due to their high charge density and preferential binding to negatively charged sites, such as phosphate groups. This facilitates selective bond cleavage and enhances fragmentation efficiency, resulting in distinct and interpretable fragment patterns. However, using non-volatile salts, such as sodium acetate, in post-column processing increases mass spectrometer contamination and background noise. OAciD MS/MS reduces the reliance on sodium adduct ions, enabling the analysis of a broader range of lipids with diverse chemical properties. Second, the C=C position can be easily annotated, even when isomers co-elute at the MS[1] level, because product ions related to the double bond are limited. For example, using EAD MS/MS requires fingerprinting the entire MS/MS spectrum for lipid annotation (Fig. S6c), whereas OAciD MS/MS simplifies the quantification of individual lipid molecules using product ions, even when isomers are not baseline-separated by LC. In our previous study, calibration curves for both EAD and OAD were assessed using a standard mixture of PC 16:0/18:1(Δ9) and PC 16:0/18:1(Δ11)[40]. Variation in their concentrations resulted in contaminated EAD MS/MS spectra for the 3:1 ratio of PC 16:0/18:1(Δ9) and PC 16:0/18:1(Δ11), leading to inconsistent annotation across analytical replicates. This contamination often results in the failure to detect both isomers. In contrast, OAD MS/MS spectra of PC 16:0/18:1(Δ9) and PC 16:0/18:1(Δ11) remained completely free of contamination in the product ion spectra.

Although the OAD MS/MS fragmentation mechanism has been demonstrated for lipidomics, reports on its application to hydrophilic metabolites remain sparse. Unlike conventional MS/MS, the OAD MS/MS system detects O- and $O_2$-attached precursor ions, with fragment ions formed by oxidation at the C=C position. While microwave discharge of ultrapure $H_2O$ produces H·, no fragment ions specifically attributable to H·-mediated cleavage of carbon-carbon single bonds were observed[36]. Consistent with previous reports, H· did not contribute to covalent bond dissociation in OAD MS/MS system. Instead, fragment ions of fatty acyl chains at the C=C position are induced by oxidation with O and OH·. The probable fragmentation pathway of OAD via O and OH· has been proposed based on density functional theory calculations, although some minor fragments have been modified during instrument development[36]. For instance, neutral losses of 96.13 Da (between Δ10 and Δ11 positions) and 138.14 Da (between Δ8 and Δ9 positions) were observed for the FA 18:1(Δ9) chain. The 96.13 Da loss occurs when OH· adds to the Δ10 position, initiating radical-induced carbon-carbon bond cleavage, which subsequently generates an enol group-containing fragment, whereas the 138.14 Da loss is initiated by O addition at the Δ9 position, leading to cleavage between Δ8 and Δ9 positions (Fig. 1a)[38]. These major fragments were detected independently of lipid subclass and MS polarity and remained detectable even at high collision energies, provided that the charge remained on the fragment containing the fatty acyl chains with double bonds.

Additionally, CID patterns and optimal collision energy were compared between Ar and $H_2O$ to achieve simultaneous fragmentation via CID and OAD. The partial pressure of Ar in the collision cell was 0.2 Pa during CID, whereas under OAD conditions, the partial pressures were 0.2 Pa for $H_2O$, and 0.04 Pa each for Ar and hydrogen, respectively, due to the specifications of the mass spectrometer (see Methods for details). Although the current OAciD system does not permit selective fragmentation solely dependent on $H_2O$, $H_2O$ vapor remains the primary collision gas. Lipid fragmentation in CID is influenced by the mass of the collision gas, as it affects the distribution of the collision energy between the lipid ion and the gas molecule. While the total collision energy is governed by the acceleration voltage applied to the mass spectrometer, importantly, the energy available for molecular fragmentation depends on the relative masses of the lipid ion and the collision gas. The use of a heavier gas, such as Ar (39.96 Da), results in greater energy transfer to the lipid ion upon collision. However, with a lighter gas, such as $H_2O$ (18.01 Da), a smaller fraction of the total collision energy is imparted to the lipid ion, necessitating a higher initial energy to achieve comparable fragmentation. In the present study, $H_2O$ required ~10 eV more collision

energy than Ar to produce similar lipid fragmentation patterns. The higher collision energy of $H_2O$ compensates for its lower energy-transfer efficiency, ensuring sufficient energy delivery to the lipid ion for bond cleavage. These differences highlight the importance of selecting an appropriate collision gas based on its mass when optimizing CID conditions for lipidomics.

Several brain morphologies, including cortical thinning, Purkinje and granule cell loss in the anterior cerebellar lobe, neuronal loss and/or decreased neuronal density in the hippocampus, are altered by aging[50–52]. Recently, marmoset brain databases, such as two- and three-dimensional brain atlases[53], the prefrontal cortex connectome[54], and in situ hybridization gene atlases[45,46], have been developed. However, research on the metabolome of mammalian brains is still in its early stages[44]. Brain lipidomics, with its structural depth and spatiotemporal localization, offers potential for characterizing altered lipid metabolism in diseases[55]. Previous studies using untargeted lipidomics in mice have revealed relationships between specific lipid metabolisms and aging, such as BMP and sulfonolipids in various organs and glycolipids in the kidney[56]. Although the lipid metabolites related to aging were not identified in this study due to the small sample size, lipid localization was assessed at a deep structural level, including C=C positions. Hierarchical clustering analysis revealed that localization patterns were primarily dependent on fatty acyl chain structures, including chain length, number of double bonds, and positions. Lipid metabolism is organized into layers of de novo synthesis of lipid subclasses (Kennedy pathway) and remodeling of fatty acyl chains (Lands cycle). A previous study suggested that n-3 very long-chain PUFA and DHA (22:6, n-3) are not incorporated into phospholipids via the same metabolic pathways[40]. Thus, the complete mechanisms of de novo synthesis and remodeling pathways in phospholipids remain unclear. While we are just beginning to investigate the localization patterns of individual lipid molecules in the brain, structural lipidomics will provide new insights into complex lipid metabolism. In addition to its structural depth, the untargeted lipidomics method and MS-DIAL 5 software characterized minor lipid subclasses, including AHexCer and MGDG, despite applying dual CID and OAD modes with $H_2O$ vapor. This study demonstrated the potential of OAciD MS/MS for next-generation lipidomics by analyzing C=C positions while detecting minor lipids in the marmoset brain.

Despite the enhanced structural depth achieved through the fragmentation method, challenges remain in the annotation and quantification of constituent molecules. The fragmentation efficiency at C=C positions is less than 5% of the precursor ion, making the dynamic range of these fragment ions too narrow for effective annotation. Further development in hardware, such as data acquisition modes, and software, such as MS-DIAL, is needed to integrate data with both low and high collision energy settings, allowing simultaneous detection of fragments related to fatty acyl composition and C=C position. This would improve deep annotation, especially for Cer and DG, whose C=C positions are difficult to annotate at high energies. Additionally, since the current OAciD system cannot determine the sn-position of fatty acyl chains, an alternative LC-MS technique, such as a two-step workflow combining OzID and COzID, is required for advanced structural lipidomics[57]. From a pretreatment perspective, enriching specific lipid subclasses is necessary to maximize the dynamic range. We have recently developed a solid-phase enrichment protocol to enhance the dynamic range, resulting in the characterization of minor lipid subclasses, such as very long-chain PUFA-containing Cer, HexCers, and dihexosylceramides (Hex2Cers) in testis, mono- and dihexosyl monoacylglycerols in feces, and acetylated and glycolylated derivatives of gangliosides using the conventional CID MS/MS system[58]. This protocol has contributed to improved lipid annotations. Furthermore, low-flow LC (<10 μL min$^{-1}$) can accelerate high-sensitivity lipidomics, although the sensitivity of C=C position-related fragment ions in dual-MS/MS spectra in negative ion mode remains a challenge. Despite these technical hurdles, this next-generation lipidomics approach could shed light on complex lipid metabolism, including the remodeling of fatty acyl chains.

## Methods

### Chemicals and reagents

Reagent grade chloroform ($CHCl_3$), 1 mol $L^{-1}$ ammonium acetate solution for high-performance LC, acetonitrile (MeCN), methanol (MeOH), 2-propanol (IPA), and ultrapure $H_2O$ for quadrupole time-of-flight MS (QTOFMS) were obtained from FUJIFILM Wako Pure Chemical Corp. (Osaka, Japan). Individual lipid synthetic standards, EquiSPLASH containing equal concentrations of deuterium-labeled lipid standard (each, 100 $\mu$g m$L^{-1}$), and UltimateSPLASH containing 69 deuterium-labeled synthetic lipids at various concentrations were purchased from Avanti Polar Lipids Inc. (Alabaster, AL, USA).

### Animals

The experimental protocols for the animal experiments were approved by RIKEN's Wako Animal Experiments Committee (W2022-2-014). Four young (1.5–1.9 years old, 3 males and 1 female) and four old (14.4–16.3 years old, all males) common marmosets were purchased from the RIKEN Research Resources Division and CLEA Japan, Inc. (Tokyo, Japan), respectively (Supplementary Data 4). After euthanasia with an overdose of ketamine (50 mg k$g^{-1}$, i.m.), the medulla, cerebellum, and midbrain were sequentially removed from the dissected brain, and the remaining brain was divided at the midline. Hippocampus and frontal lobe were collected from the right brain. Brain tissues were harvested and immediately frozen after dissection and stored at −80 °C until lipid extraction.

### Liquid-liquid extraction

The Bligh and Dyer method was used as previous method[59,60]. Marmoset brain sections were lyophilized overnight in the dark. After the brain tissue was crushed using a ball mill, an average volume of 5 mg was transferred to a clean tube. Samples were mixed with 1000 $\mu$L of MeOH/$CHCl_3$/$H_2O$ (10:4:4, v/v/v). Lipids were extracted using a vortex mixer for 1 min and then ultrasonicated for 5 min. The solution was centrifuged at 16,000 × $g$ for 5 min at 4 °C, and 700 $\mu$L of the supernatant was transferred to a clean tube. The supernatant was mixed with 235 $\mu$L of $CHCl_3$ and 155 $\mu$L of $H_2O$ using a vortex mixer for 1 min. Both layers were collected after centrifugation at 16,000 × $g$ for 15 min at 4 °C. Lipid extracts were dried using a centrifuge evaporator. They were dissolved with MeOH containing 200 times diluted EquiSPLASH mixture (each, 500 ng m$L^{-1}$).

### High-throughput analysis for method optimization

The LC/QTOFMS system was composed of a Nexera and an LCMS-9030 equipped with an OAD RADICAL SOURCE I (Shimadzu Corp., Kyoto, Japan). A high-throughput analytical system was used to efficiently optimize the method without causing an MS shift or change in sensitivity owing to the long period of operation[61]. The LC conditions were as follows: injection volume, 1 $\mu$L; mobile phase, MeCN/MeOH/$H_2O$ (1:1:3, v/v/v) (A) and MeCN/IPA (1:9, v/v) (B) (both contained 10 nM of ethylenediaminetetraacetic acid (EDTA) and 5 mM of ammonium acetate); flow rate, 600 $\mu$L mi$n^{-1}$; column; Unison UK-C18 MF (50 × 2.0 mm, 3 $\mu$m, Imtakt Corp., Kyoto, Japan); gradient, 0.1% (B) (0.1 min), 0.1–15% (B) (0.1 min), 15–30% (B) (0.9 min), 30–48% (B) (0.3 min), 48–82% (B) (4.2 min), 82–99.9% (B) (1.3 min), 99.9% (B) (0.2 min), 99.9–0.1% (B) (0.1 min), 0.1% (B) (1.4 min); column oven temperature, 65 °C. The column eluent was mixed with 10–50 $\mu$L mi$n^{-1}$ of 50% MeOH containing 200 $\mu$M sodium acetate post-column when forming the sodium adduct ion (Figure S1a). MS conditions were as follows: nebulizing gas flow, 3.0 L mi$n^{-1}$; heating gas flow, 10.0 L mi$n^{-1}$; drying gas flow, 10.0 L mi$n^{-1}$; interface temperature, 300 °C; DL temperature, 250 °C; heat block temperature, 400 °C; CID gas, 17 kPa (using $H_2O$ and its radicals) or 230 kPa (using Ar gas); TOF range; $m/z$ 75–1250; event time; 200 ms (MS scan) or 100 ms (MS/MS); interface voltage, 4 kV (positive ion mode) or −4 kV (negative ion mode); Q1 resolution, 4 Da. The partial pressure of Ar in the collision cell was 0.2 Pa when the inlet pressure was set to 230 kPa. Under OAD conditions, the partial pressure of $H_2O$ was also 0.2 Pa. Based on the LCMS-9030 system specifications, the partial pressure of Ar was 0.04 Pa when the inlet pressure was set

to 17 kPa (low setting). Additionally, hydrogen gas was introduced at 0.04 Pa to reduce oxidation by O/OH radicals on the quadrupole electrodes, preventing insulating layers and ensuring stable potential. The precursor ions of the product ion scan used to investigate the fragmentation efficiency between positive and negative ion modes and to optimize the collision energy are summarized in Supplementary Data 1.

### Lipidome analysis of the marmoset brains

The LC conditions were as follows: injection volume, 2 $\mu$L; mobile phase, MeCN/MeOH/$H_2O$ (1:1:3, v/v/v) (A) and MeCN/IPA (1:9, v/v) (B) (both contained 10 nM of EDTA and 5 mM of ammonium acetate); flow rate, 300 $\mu$L mi$n^{-1}$; column; Unison UK-C18 MF (50 × 2.0 mm, 3 $\mu$m, Imtakt Corp., Kyoto, Japan); gradient, 0.5% (B) (1 min), 0.5–40% (B) (4 min), 40–64% (B) (2.5 min), 64–71% (B) (4.5 min), 71–82.5% (B) (0.5 min), 82.5–85% (B) (6.5 min), 85–99% (B) (1 min), 99% (B) (2 min), 99–0.5% (B) (0.1 min), 0.5% (B) (2.9 min); column oven temperature, 45 °C. LCMS-9030 conditions were the same as those used for method optimization.

Structural analysis of PC 16:0_16:1 and PC 16:0_18:1 was performed using a ZenoTOF 7600 (SCIEX, Framingham, MA, USA). ZenoTOF 7600 conditions were used as previous method[40]. MS parameters were as follows: ion source gas 1, 40 psi; ion source gas 2, 80 psi; curtain gas, 30 psi; CAD gas, 7; temperature, 250 °C; polarity, positive ion mode; spray voltage, 5500 V; declustering potential, 80 V; collision energy, 10 V; scan range; $m/z$ 75–1250 (MS$^1$) and 100–1250 (MS/MS); accumulation time, 200 ms (MS$^1$) and 100 ms (MS/MS); time bins to sum, 4 (MS$^1$) and 8 (MS/MS); electron beam current, 7000 nA; zeno threshold, 1,000,000 cps; kinetic energy, 14 and 18 eV; EAD RF, 150 Da; reaction time, 98 ms.

### Data analysis

In the method development, peak area was calculated using LabSolutions Insight LCMS (Shimadzu). For the lipidome analysis of marmoset brains, lipid molecules were annotated using MS-DIAL version 5.1. Representative parameters were as follows: minimum peak height for peak picking, 100; retention time tolerance for peak alignment, 0.15 min; MS$^1$ tolerance, 0.01 Da; MS$^2$ tolerance, 0.025 Da. The other data processing parameters and adduct ions used for each lipid subclass are listed in Supplementary Data 4. The annotated results were manually curated using the retention times and MS/MS spectra of the peaks in both positive and negative ion modes. The data matrix was normalized based on the dry weight of the marmoset brain tissue and the ion abundance of the EquiSPLASH mixture. The lipidome results are summarized in Supplementary Data 4.

### Lipid nomenclature and quantification

The nomenclature used in this manuscript is described in accordance with the Lipidomics Standards Initiative[62]. For example, PC 16:0_18:1 indicates that the individual fatty acyl chains were determined by MS/MS. The detailed C=C position is described as PC 16:0_18:1(Δ9) when fragment ions derived from the C=C positions are obtained by OAciD. In contrast, the type of quantification corresponded to level 3 (lipid subclass other than analyte or no co-ionization of analyte and internal standard) of the Lipidomics Standards Initiative International Guidelines. The corresponding internal standards used to quantify each lipid subclass are summarized in Supplementary Data 4. The quantitative performance characteristics of the RPLC/MS system were discussed in a previous report[58].

### Data availability

All raw MS data are available on the RIKEN DROPMet website (https://prime.psc.riken.jp/menta.cgi/prime/drop_index) under index number DM0066 and in the MB-POST repository (https://repository.massbank.jp/) under index number MPST000010. The lipidomics data extracted from MS raw data are available as Supplementary Data 1, 2, 3, and 4.

### Code availability

MS-DIAL source code is available at https://github.com/systemsomicslab/MsdialWorkbench.

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

## Acknowledgements

We thank the staff of the Research Resources Division, RIKEN Center for Brain Science, for the management and breeding of some of the marmosets. We are also grateful to Dr. Daiki Asakawa (National Institute of Advanced Industrial Science and Technology) for the technical advice on $H_2O$ vapor fragmentation. This research was supported by the Japan Agency for Medical Research and Development (AMED) under Brain Mapping by Integrated Neurotechnologies for Disease Studies (Brain/MINDS) (JP15dm0207001 to H.O., H.K., and H. Tsugawa, and JP23wm0625001 to H.O.), AMED for Infectious Diseases Research and Infrastructure (JP25wm0325071, H. Tsugawa), the Japan Science and Technology Agency (JST) Exploratory Research for Advanced Technology (ERATO) (JPMJER2101 to H. Tsugawa), JST FOREST program (JPMJFR230H to H. Tsugawa), JST NBDC (JPMJND2305 to H. Tsugawa), the JSPS KAKENHI (24K02011, 24H00043, 24H00392, 24K21269 to H. Tsugawa), Environmental Restoration and Conservation Agency Cross-ministerial Strategic Innovation Promotion Program (JPJ012290 to H. Tsugawa) and Technologically Advanced research through Marriage of Agriculture and engineering as Groundbreaking Organization (TAMAGO to H. Tsugawa).

## Author contributions

H. Takeda, H.K., and H. Tsugawa designed the study. M.O. and H. Takahashi developed the hardware for the OAD MS/MS system. B.B. and H. Tsugawa developed MS-DIAL software. H. Takeda and M.O. optimized the method, analyzed the lipids, and performed peak annotation and quantification. H.O. and H.K. prepared the animals, and N.K. dissected them. H. Takeda and H. Tsugawa prepared the manuscript. H. Takeda, B.B., and H. Tsugawa prepared the figures. All the authors approved the final version of the manuscript.

## Competing interests

M.O. and H. Takahashi are research scientists at Shimadzu, Inc., Japan. All the other authors declare that they have no competing interests.
