## [Transparent Peer Review file · Communications Chemistry]

Dual fragmentation via collision-induced and oxygen attachment dissociations using water and its radicals for C=C position-resolved lipidomics

Corresponding Author: Dr Hiroshi Tsugawa

Version 0:

Reviewer comments:

Reviewer #1

(Remarks to the Author)

The authors propose a convenient solution for acquiring CID and OAD specific fragment ions in a single MS measurement. The approach takes advantage of residual H₂O vapor that is present in the system as a matter of the characteristic radical generation process for OAD. The resulting dual fragmentation process termed OAciD promises a significant acceleration of OAD-based workflows. Previously, CID measurements for lipid identification at the molecular species level had to be acquired in addition to conventional OAD. Notably, the authors demonstrate the applicability of the approach in the negative ion mode in principle (see major comment 2). This is crucial for the analysis of several lipid subclasses that ionize better in negative ion mode, such as PI and PG.

The study is overall performed with technical rigor and principally sound. However, there are a few concerns.

Major comments

1. The relative quantification of the two double bond isomers ($\Delta 7$ vs $\Delta 9$) is questionable. This comment pertains to Figure 4a, Figure 4d, and line 272: In Figure 4a, please clearly highlight the fragments used for relative quantification in the provided OAciD spectra for the two isomers, i.e. the ones at m/z 636.42 and m/z 664.45. In the current reference spectrum of the $\Delta 9$ version of PC 16:0_16:1 in Figure 4a, it seems that there is also a reference intensity present at m/z 636.42, which is characteristic for $\Delta 7$. From the provided reference spectra, it seems that both isomers yield a fragment at m/z 636.42. If this is the case, please take correction of potentially shared m/z values into account for the relative quantification used for creating Figure 4d, and describe in more detail how you derived these values. Moreover, it is surprising that $\Delta 7$ is present in such high amounts, because typically 16:1 ($\Delta 9$) is prevalently detected in mammalian cells. A fragment contribution of 16:1 ($\Delta 9$) to m/z 636.42 would promote such a result. If the reference fragment from 16:1 ($\Delta 9$) can be clearly distinguished from m/z 636.42, please demonstrate that more clearly in the spectrum.
2. Figure S5: Please elaborate on the reason of uncharacterized spectra, especially for lipid classes that would particularly benefit from analysis in negative ion mode. Explain the discrepancy between Figure 2b and S5: In Figure 2b, e.g., PG was annotated in several measurements and double bonds were identified, whereas in Figure S5, it appears that the method does not work for PG. If the reason derives from an incomplete spectral library, we recommend to extend it accordingly.
3. Figure 2d and lines 194, 195: It is unclear how MS-DIAL obtained annotations at the MSL and MSL+DB level for all spectra that are identified only at SL level in Figure 2c. Did MS-DIAL use OAD fragments for MSL annotations in this case? Or did MS-DIAL obtain this information from other experiments (by e.g., matching retention times from previous experiments), where you used optimized collision energies for CID, as it was the case for previous OAD setups? In the latter case, please describe these experiments in detail also. Or did you manually provide this information to MS-DIAL? In any case, this information must be provided somewhere in the manuscript.
4. Line 60: 'without requiring derivatization or specialized equipment.' – OAD is a proprietary fragmentation method by Shimadzu and only works with a Shimadzu QTOF system equipped with an OAD RADICAL SOURCE I. Similarly, EAD also only works with specialized MS instruments. Accordingly, it is misleading to claim that these methods do NOT require 'specialized equipment'. Please remove this claim

Minor comments

1. Please provide your data at an independent public repository, such as GNPS.
2. Line 82: Please cite the latest version of the MS-DIAL 5 publication that has been published in Nature Communications recently.
3. Ambiguous meaning at lines 88-89: "... in the negative ion mode, a domain that has not been thoroughly explored for lipids." This phrasing gives the impression that this study is the first one using negative ion mode for lipids. Please rephrase. Suggestion: "... OAD-MS/MS in negative ion mode, which was used for the first time for this polarity."
4. Figure 1d: Several displayed fragment m/z values seem to be incorrect, e.g. 711.57 > 467.37 in blue should probably be 570.55 > 467.37. The same is true for Figure S3, to give another example. Please thoroughly check and correct these m/z transitions. In addition, please specify what the 'Relative peak area' (y-axis) is relative to.
5. Figure S3: Why is there no 'CID fragmentation from fatty acyl composition' line plot for PI? Figure 1c indicates that these fragments were measured. Please show this information for all lipid classes where it is available.
6. Figure 2b: Why is there no double bond position assignment for Cer? Double bonds were identified for similar Cer compounds at similar concentration ranges for the USO standard mixture (Fig 2c,d).
7. Figure 2c,d: There is no apparent reason why there are no MS2 spectra of CE, LPG and LPI available, since PG and PI are reported with the same measurement conditions. Please elaborate on this point. In addition, please clarify whether double bonds were assigned for the sphingoid base or only to the N-acyl chain for Cer and SM.
8. Line 190: 'the annotation results of SM, PE, and TG molecules became species level (SL)' – this is inconsistent with Figure 2c, where all TG molecules are reported at least at MSL level.
9. Optimization of fragmentation rules for Cer at lines 214-216: The EquiSPLASH mixture contains only one Cer compound (Cer 18:1;O2-d7/15:0). It is not apparent why the UltimateSPLASH ONE mixture was not used for this purpose. This standard mixture contains five different Cer molecules, which would give more confidence in the derived decision tree that was extrapolated to sulfatides.
10. Lines 247-249 and 293-296: Did you intend to say: "OAcid-MS/MS is advantageous compared to EAD in a situation of two co-eluting C=C isomers, because..."? In addition, the claim that both coeluting isomers cannot be identified by EAD requires stronger evidence. The two spectra appear quite clearly distinguishable by the two highlighted fragments in the referenced Figure S6c, where it is shown that fragments produced by the McLafferty rearrangement allow for the assignment of the double bond positions. And from the intensity of these fragments, it is apparent that they can be distinguished. Please either tone down this claim or provide stronger evidence, e.g., by performing a dilution series between two such isomeric standards to determine the ratio at which the lower abundant isomer is still detectable with either method.
11. Line 407: Please change "detailed position" to "detailed C=C position" or "detailed double bond position" to avoid confusion with sn -position assignment.
12. Line 437: 'Circles represent the linear range of the MS1 peak with $R^2 > 0.9800$ (Table S1).'; Shouldn't this be Table S2?
13. Suggestion for line 437: Please change "The closed circles" to "The fully filled circles ..."

Reviewer #2

(Remarks to the Author)

I co-reviewed this manuscript with one of the reviewers who provided the listed reports. This is part of the Communications Chemistry initiative to facilitate training in peer review and to provide appropriate recognition for Early Career Researchers who co-review manuscripts.

Reviewer #3

(Remarks to the Author)

This manuscript describes the use of combined collision induced dissociation (CID) and oxygen attachment dissociation (OAD) tandem mass spectrometry (MS/MS) methods to enable high confidence structural identification in lipidomics workflows. Building off prior work, this combined approach is novel and is also more readily applied to negative ion mode lipid analysis. A wide variety of lipid classes, fatty acyl tails, and double bond positions are identified. The work is nicely contextualized by analyzing complex extracts from marmoset brain. While sufficient replicates are not analyzed to draw biological conclusions comparing young and old brains, the lipid observations are nicely contextualized. The figures, though complex and a lot to digest, are informative and clear. There are a few instances where comparisons and contextualization to existing literature would be relevant (e.g., references in the Introduction, comparisons to MSI). Please see below for specific questions and comments:

1. Several references should probably be added:

- a. OzID: The Blanksby lab pioneered OzID and has published 10-20 papers on this technique, but their work is not referenced. In particular, several recent reports use OzID in a similar manner to the workflow proposed in this manuscript: <https://doi.org/10.1002/anie.202316793> and <https://doi.org/10.1038/s41467-023-39617-9>
- b. Ion/ion reactions: Gas-phase ion/ion reactions have also been used to reveal lipid sn-, double bond, and stereochemistry positions, and are worth mentioning (mainly from the McLuckey and Prentice labs):
- i. <https://doi.org/10.1021/acs.analchem.0c02350>
 - ii. <https://doi.org/10.1002/anse.202300063>
 - iii. <https://doi.org/10.1021/acs.analchem.3c03804>
 - iv. <https://doi.org/10.1021/jasms.4c00368>
 - v. <https://doi.org/10.1021/acs.analchem.8b03441>
 - vi. <https://doi.org/10.1021/acs.analchem.9b04376>
 - vii. <https://doi.org/10.1021/ac400190k>
 - viii. <https://doi.org/10.1021/acs.analchem.5b02243>
- c. EAD: The Prentice and Kane labs have also contributed to lipid EAD:
- i. <https://doi.org/10.1002/jms.3698>
 - ii. <https://doi.org/10.1016/j.ijms.2020.116338>
 - iii. <https://doi.org/10.1016/j.ijms.2022.116998>
 - iv. <https://doi.org/10.1021/acs.analchem.3c03077>
2. Nomenclature: The hyphen between "CID-MS/MS" and "OAD-MS/MS" should be removed. Hyphens in these types of abbreviations are used to denote orthogonal analytical techniques coupled together (e.g., LC-MS/MS and GC-MS/MS). CID is not being coupled to MS/MS in these instances; it is the type of MS/MS being performed. "CID MS/MS" or "MS/MS via CID" would be more appropriate terminology.
3. Could the authors describe (and perhaps show in the Supplementary Information) the mechanism for OAD? This would facilitate understanding the fragmentation discussion.
4. The authors attribute the high fragmentation efficiency of sodium adducted ions to "the strong charge bias introduced by metal ions like sodium, compared to proton or ammonium ions." What does the term "charge bias" mean and how does this affect the efficiency of the fragmentation mechanism?
5. What is the nominal pressure in the collision cell, and what is the partial pressure of water? It seems likely that the H₂O versus Ar experiments are really comparing background N₂ versus Ar gas behavior (i.e., the ions are not largely undergoing CID via collisions with H₂O). The higher mass of Ar would also support a lower collision energy, as the energy partitioning into the ion would be more efficient than with a lighter neutral target like N₂.
6. The statement that "little work has been done on the lipid profiling of mammalian brain regions" seems a bit odd (there are dozens of MALDI and DESI mass spectrometry imaging studies that have mapped lipid distributions in mammalian brains). How do the lipid abundances observed here compare to previous MSI studies?
7. It appears that positional sn- isomers are not resolved in this method (e.g., PC 16:0/18:1 versus PC 18:1/16:0), which should be mentioned.

Reviewer #4

(Remarks to the Author)

The authors present results of combining the oxygen-addition dissociation (developed by some of them) with traditional collision-induced dissociation for lipid structural characterisation. The combined fragmentation patterns, notably in negative ion mode, are successfully demonstrated to yield a richer structural assignment that utilising either method in isolation; including the identification of carbon-carbon double bonds and discrimination between potential lipid isomers. The method is deployed against a sizeable set of lipid standards and biological extracts. The experiments are largely consistent with the data acquired and data analysis has been incorporated into MS-DIAL; a widely used software suite for lipid –and other metabolite– analysis. The manuscript should be published after the authors consider the following suggestions and corrections:

- (1) The literature review and introduction are superficial. For example, the alternative and well-established methods of OzID and UVPD are mentioned but the original papers introducing these methods for lipid analysis are not cited. The papers cited for OzID are actually not comparable to the methods described herein as the ozonolysis is not conducted on mass selected ions. The UVPD citation references a combination method not the comparable UVPD of mass-selected ions. Both methods have been widely explored for positive and negative ions in a similar manner to much of the OAD discussion in this paper. The primary literature should be carefully reviewed and summarised in the context of the current study.
- (2) Similar to (1), the advantages of using CID with OzID (DOI: 10.1039/c3an01712e) and UVPD (DOI: 10.1021/jacs.7b06416) have been described before and the advantages of these combinations for full structural assignment are described in detail. In addition, a combination CID-OzID experiment (i.e., without MS3 mass-selection) has also been described (DOI: 10.1016/j.ijms.2018.05.016) that is highly relevant to the composite OAD-CID experiment outlined here. These prior studies find that composite dissociation can still resolve product ions arising from sequential CID/OzID processes, i.e., ozonolysis of CID product ions. It is unclear whether any such ions are observed here?
- (3) Pg 2, line 1. What are the energy storage materials?
- (4) Pg 2, line 45. What is meant by "description in the quality and quantity of lipids"? this is unclear.
- (5) Pg 2, line 60. The introduction of other techniques is confused as it seems to intertwine methods relying on external derivatization of lipids prior to mass spectrometry with those that use selective dissociation of underivatized lipids, namely, EAD, OzID, UVPD and OAD. This should be clarified and appropriately referenced. Similarly, the use of the term "specialised equipment" is specious here. All these techniques require specialised equipment. With the exception of OzID all have been commercialised. Commercialisation of a technology does not make it any less specialised.
- (6) Pg 3, Line 89. The statement that negative ions are "a domain that has not been thoroughly explored for lipids" is

incorrect. See any text book on lipid structural analysis (e.g., Murphy's Tandem Mass Spectrometry of Lipids) which are full of negative ion analysis.

(7) Pg 3, line 113. What is "charge bias"?

(8) Pg 9, line 305. The chemical descriptions of the radical chemistries are inaccurate. Hydroxyl radicals can add to a carbon-carbon bond with subsequent hydrogen atom abstraction (by oxygen or other scavengers) or direct elimination of hydrogen yields the new double bond.

(9) Pg 9 (and elsewhere). One of the limitations of negative ion CID fragmentation of glycerolipids has been the inability to distinguish between BMP and PG isomers. It would be interesting to see whether the competitive CID-OAD spectra provide unique assignments. How are the authors sure they are assigning one and not the other?

Version 1:

Reviewer comments:

Reviewer #1

(Remarks to the Author)

The authors satisfactorily addressed my previous concerns. The authors addressed all comments of Reviewer #4 adequately. I have no more comments.

Reviewer #2

(Remarks to the Author)

I co-reviewed this manuscript with one of the reviewers who provided the listed reports. This is part of the Communications Chemistry initiative to facilitate training in peer review and to provide appropriate recognition for Early Career Researchers who co-review manuscripts.

Reviewer #3

(Remarks to the Author)

The authors have appropriately addressed my prior concerns (original Reviewer 3) and the manuscript is suitable for publication.

Reviewer #1 (Remarks to the Author):

The authors propose a convenient solution for acquiring CID and OAD specific fragment ions in a single MS measurement. The approach takes advantage of residual H₂O vapor that is present in the system as a matter of the characteristic radical generation process for OAD. The resulting dual fragmentation process termed OAciD promises a significant acceleration of OAD-based workflows. Previously, CID measurements for lipid identification at the molecular species level had to be acquired in addition to conventional OAD. Notably, the authors demonstrate the applicability of the approach in the negative ion mode in principle (see major comment 2). This is crucial for the analysis of several lipid subclasses that ionize better in negative ion mode, such as PI and PG.

The study is overall performed with technical rigor and principally sound. However, there are a few concerns.

Thank you for your comments. Our responses are in blue, and major revisions are highlighted in yellow.

Major comments

1. The relative quantification of the two double bond isomers ($\Delta 7$ vs $\Delta 9$) is questionable. This comment pertains to Figure 4a, Figure 4d, and line 272: In Figure 4a, please clearly highlight the fragments used for relative quantification in the provided OAciD spectra for the two isomers, i.e. the ones at m/z 636.42 and m/z 664.45. In the current reference spectrum of the $\Delta 9$ version of PC 16:0_16:1 in Figure 4a, it seems that there is also a reference intensity present at m/z 636.42, which is characteristic for $\Delta 7$. From the provided reference spectra, it seems that both isomers yield a fragment at m/z 636.42. If this is the case, please take correction of potentially shared m/z values into account for the relative quantification used for creating Figure 4d, and describe in more detail how you derived these values. Moreover, it is surprising that $\Delta 7$ is present in such high amounts, because typically 16:1($\Delta 9$) is prevalently detected in mammalian cells. A fragment contribution of 16:1($\Delta 9$) to m/z 636.42 would promote such a result. If the reference fragment from 16:1($\Delta 9$) can be clearly distinguished from m/z 636.42, please demonstrate that more clearly in the spectrum.

We apologize for the lack of clarity in Figure 4a and the related descriptions. We have added **Figure S7** to clearly highlight the fragment ions associated with the $\Delta 7$ and $\Delta 9$ C=C positions. Additionally, we have provided a reviewer-only figure based on **Supplementary Figure 7F** from Takeda et al., *Nature Communications* **15**, 9903 (2024), where PC 16:0_18:1($\Delta 9$) and PC 16:0_18:1($\Delta 11$) were used as examples to illustrate OAD MS/MS fragmentation patterns. While authentic standards for PC 16:0_16:1($\Delta 9$) and PC 16:0_16:1($\Delta 7$) are unavailable, we think that the same principle can be applied. As shown in **Supplementary Figure 7F**, PC 16:0_18:1($\Delta 9$) generates m/z 664, 654, and 622 as C=C position-specific fragment ions, with m/z 664 and 622 serving as diagnostic ions. In contrast, PC 16:0_18:1($\Delta 11$) produces m/z 692, 682, and 650, confirming that these isomers do not share common C=C position-specific fragment ions between these isomers.

f. OAD-MS/MS fragmentation patterns of PC 16:0/18:1(9) and PC 16:0/18:1(11)

Zoom in for the above spectra: PC 16:0/18:1(9) 500 nM vs PC 16:0/18:1(11) 500 nM

Supplementary Figure 7F from Takeda et al., Nature Communications 15, 9903 (2024)

To clarify our results, we created **Figure S7**, which presents four possible structural isomers: PC 16:0_16:1(Δ 9), PC 16:0_16:1(Δ 7), PC 14:0_18:1(Δ 9), and PC 14:0_18:1(Δ 11), all of which may co-elute. We also included the MS/MS spectrum from the marmoset sample with the highest peak intensity (20240130_Marmo9_young_hippocampus_OAD_POS_1_003). The data confirm that PC 14:0_18:1 isomers

contribute negligibly, indicating that the diagnostic fragment ions for C=C positions primarily originate from PC 16:0_16:1(Δ 9) and PC 16:0_16:1(Δ 7) (m/z 594 and 636 for Δ 7, and m/z 622 and 664 for Δ 9). No other structurally relevant fragments were detected, supporting the reliability of using m/z 664 and 636 for relative quantification.

Newly created Figure S7

Our previous study using authentic PC 16:0_18:1(Δ 9) and PC 16:0_18:1(Δ 11) demonstrated that C=C position-specific fragment ion intensities remain relatively consistent across different C=C positions, enabling the estimation of isomer ratio in biological samples. However, the presence of other potential isomers cannot be entirely ruled out, and we did not validate our results with authentic PC 16:0_16:1 standards. Therefore, we have clarified in the revised manuscript that the reported PC 16:0_16:1(Δ 9) to PC 16:0_16:1(Δ 7) ratio should be considered a semi-quantitative estimation (lines 321–326).

"According to our previous study, the C=C position-specific fragmentation efficiency between PC 16:0_18:1(Δ 9) and PC 16:0_18:1(Δ 11) was nearly equal, allowing the relative abundance of these lipid isomers to be estimated by MS/MS peak heights. Thus, the product ion peak heights of m/z 636.42 and m/z 664.45, which were not interfered with co-eluting isomers, were used to calculate the ratio of PC 16:0_16:1(Δ 7) to PC 16:0_16:1(Δ 9) (Figure S7). However, these ratios should be considered semi-quantitative due to the absence of authentic standards for these isomers."

Furthermore, we identified a bug in the *in-silico* MS/MS spectral library generation program of MS-DIAL 5, which

resulted in complex and potentially misleading reference spectra in **Figure 4a**. We have corrected the program, and we believe that the revised **Figure 4a**, along with the newly created **Figure S7**, more accurately represents our results.

2. Figure S5: Please elaborate on the reason of uncharacterized spectra, especially for lipid classes that would particularly benefit from analysis in negative ion mode. Explain the discrepancy between Figure 2b and S5: In Figure 2b, e.g., PG was annotated in several measurements and double bonds were identified, whereas in Figure S5, it appears that the method does not work for PG. If the reason derives from an incomplete spectral library, we recommend to extend it accordingly.

Thank you for your insightful comment. During validation of the automatic annotation pipeline in MS-DIAL, we identified a critical bug affecting OAD MS/MS annotation, particularly in negative ion mode. The MS-DIAL workflow checks for lipid subclass and fatty acyl-specific ions for molecular species-level (MSL) annotation and subsequent C=C position assignment (or SN1/SN2-specific diagnostic ion searches in the case of EAD MS/MS). However, we discovered that the OAD MS/MS pipeline erroneously used the EAD MS/MS pipeline, which was primarily developed for positive ion mode. As a result, negative ion spectra were not processed accurately, leading to annotation failure. This issue arose due to insufficient oversight in the open-source development of MS-DIAL, where contributions from multiple developers affected code consistency. After correcting the algorithms, we reanalyzed the OAcID MS/MS data and updated Figures 2c/d and S5. The revised results indicate that PG, PI, and PS were successfully annotated at the MSL+DB level in high-concentration regions, confirming that OAD MS/MS is applicable to these lipid subclasses. The previous discrepancy between Figure 2b and Figure S5 was due to the annotation pipeline error rather than a limitation of the method. Additionally, PI and SM remained “uncharacterized” in negative ion mode (Figure S5c) because lipid subclass- or fatty acid-specific product ions were not detected at 30 eV (see Figure S2). To address this, we tested MS-DIAL under conditions where MSL annotation was manually assigned (Figure 2d), confirming that appropriate collision energy conditions are required for targeted lipid subclasses, while double bond annotations remain feasible when MSL annotation is provided (Figures S5b and S5d). We sincerely appreciate your feedback, which helped us identify and resolve this critical issue. The revised manuscript, along with the updated Figure S5, now clearly presents the advantages and limitations of the proposed technique.

3. Figure 2d and lines 194, 195: It is unclear how MS-DIAL obtained annotations at the MSL and MSL+DB level for all spectra that are identified only at SL level in Figure 2c. Did MS-DIAL use OAD fragments for MSL annotations in this case? Or did MS-DIAL obtain this information from other experiments (by e.g., matching retention times from previous experiments), where you used optimized collision energies for CID, as it was the case for previous OAD setups? In the latter case, please describe these experiments in detail also. Or did you manually provide this information to MS-DIAL? In any case, this information must be provided somewhere in the manuscript.

Thank you for your comments. To clarify the differences in evaluation methods between Figure 2c and Figure 2d, we have added the following sentences to the revised main text (lines 231–238):

"Thus, we further evaluated the MS-DIAL 5 program in cases where MSL annotations for all lipid subclasses were pre-defined before the C=C position determination to examine only the OAD spectrum (Figures 2d and S5b). For example, when annotating the spectrum of PC_d5 17:0_16:1(Δ^9), the peak was manually assigned to PC_d5 17:0_16:1 in MS-DIAL, ensuring that molecular species-level (MSL) annotation was established before initiating C=C position searches. This approach is executable in the current version of MS-DIAL through a user-defined text library containing compound names, precursor m/z values, and retention time information. The results demonstrated that MS-DIAL 5 accurately annotated the C=C positions of phospholipids and SM when the acyl chain information was provided."

This means that, as shown in Figure 2d, the MSL annotation was assigned based on prior knowledge rather than inferred solely from the OAcID MS/MS spectrum. However, Figure 2c illustrates MS-DIAL's attempts to characterize lipid molecules without predefined information.

4. Line 60: *'without requiring derivatization or specialized equipment.'* – OAD is a proprietary fragmentation method by Shimadzu and only works with a Shimadzu QTOF system equipped with an OAD RADICAL SOURCE I. Similarly, EAD also only works with specialized MS instruments. Accordingly, it is misleading to claim that these methods do NOT require 'specialized equipment'. Please remove this claim

Thank you for your comments. A similar concern was raised by Reviewer #4 in Comment #5. To prevent any potential misunderstanding, we have removed the phrase "or specialized equipment" from the manuscript.

Minor comments

1. Please provide your data at an independent public repository, such as GNPS.

Thank you for the suggestion. A new metabolomics data repository, MB-POST, was recently launched in Japan. Accordingly, we have uploaded our LC-MS/MS data to the MB-POST repository under the index ID MPST000010. The data will be made available upon the official publication of the manuscript. You can access the repository at <https://repository.massbank.jp/preview/202190196767a773f5e7c50> under the PIN code of "4099".

2. Line 82: Please cite the latest version of the MS-DIAL 5 publication that has been published in Nature Communications recently.

Thank you. We have updated.

3. Ambiguous meaning at lines 88-89: "... in the negative ion mode, a domain that has not been thoroughly explored for lipids." This phrasing gives the impression that this study is the first one using negative ion mode for lipids. Please rephrase. Suggestion: "... OAD-MS/MS in negative ion mode, which was used for the first time for this

polarity.”

Thank you for the suggestion. We have revised the sentence to "... OAD MS/MS in negative ion mode, which was used for the first time for this polarity." to ensure clarity and avoid misleading implications (lines 119–121).

4. Figure 1d: Several displayed fragment m/z values seem to be incorrect, e.g. $711.57 > 467.37$ in blue should probably be $570.55 > 467.37$. The same is true for Figure S3, to give another example. Please thoroughly check and correct these m/z transitions. In addition, please specify what the ‘Relative peak area’ (y-axis) is relative to.

Thank you for your careful review of the manuscript. The correct description should be " m/z $755.56 > 467.37$ ". Initially, PE 15:0_18:1(d7) (precursor m/z 711.57) was included in Figure 1d, but it was later replaced with PS 15:0_18:1(d7) (m/z 755.56). However, we inadvertently failed to update the label from "711.57" to "755.56," which has now been corrected. Regarding the " $570.55 > 467.37$ " transition, we understand the potential confusion. As you correctly noted, m/z 467.37 originates from the neutral loss of the polar head group (m/z 570.55), followed by C=C position-specific fragmentation. However, since m/z 467.37 is ultimately a product ion derived from the precursor m/z 755.56, we maintain that the appropriate notation is " m/z $755.56 > 467.37$ " rather than referencing the intermediate step. The figure legend provides a sufficient explanation of this description. The relative peak area (%) on the y-axis is defined as follows:

"Relative peak areas (%) denote the ion intensities in the product ion spectrum, with the most intense peak set to 100%."

We updated the figure legends for Figure 1d and Figure S3 to reflect this clarification.

5. Figure S3: Why is there no ‘CID fragmentation from fatty acyl composition’ line plot for PI? Figure 1c indicates that these fragments were measured. Please show this information for all lipid classes where it is available.

We are not entirely certain whether we have fully understood your comment; however, we will address two possible interpretations:

Difference between Figure 1c and Figure S3a: Figure 1c presents negative ion mode data, whereas Figure S3a presents the positive ion mode data. Under our measurement conditions, PI-derived acyl chain fragments were rarely detected in positive ion mode (below the detection limit). Therefore, the acyl chain-derived product ion information (green) is not included in Figure S3a.

Regarding Figure S3b: In negative ion mode, acyl chain-derived fragment ions were detected at very high intensity for all phospholipids including PI. These fragment ions are often more than ten times more intense than the OAD-derived C=C position-specific fragments. Since the primary goal of Figure S3 is to emphasize the collision energy dependency of double-bond-specific fragment ions, including acyl chain-derived ions (which exhibit vastly different intensity ranges), it would obscure this trend. Therefore, we intentionally excluded fatty acyl fragment ions from

Figure S3b. The collision energy dependence of fatty acyl fragment ions is presented separately in Figure S2.

6. *Figure 2b: Why is there no double bond position assignment for Cer? Double bonds were identified for similar Cer compounds at similar concentration ranges for the USO standard mixture (Fig 2c,d).*

Thank you for your comments. Your comment highlights the need for a more detailed explanation regarding C=C position assignment in sphingolipids in this study. Under our experimental conditions, C=C position-specific fragment ions for the sphingoid base backbone, particularly for the $\Delta 4$ double bond, were not detected. As shown in **Figure S3a**, fragmentation efficiency at the C=C position in sphingoid bases decreased with increasing collision energy, and no fragmentation was observed above 15 eV. This contrasts with phospholipids, where neutral loss fragments related to C=C positions were more consistently observed (blue bar in **Figure S3a**). As noted in the method limitations, integrating low- and high-collision-energy settings is crucial for accurate C=C position determination in ceramides. Additionally, fragmentation efficiency for other potential C=C positions in sphingoid bases, such as $\Delta 8$ or $\Delta 14$, was not evaluated in this study. In contrast, validation using the UltimateSPLASH standard mixture confirmed that C=C position-specific fragment ions from the *N*-acyl chain were detectable. Therefore, in MS-DIAL's validation using UltimateSPLASH, when the *N*-acyl chain's C=C position was assigned, the annotation was labeled as "MSL+DB." To clarify this, we have updated the figure legends for Figures 2c and 2d with the following explanation:

"In this study, fragment ions related to sphingobase $\Delta 4$ C=C were not detected under the applied collision energy settings. However, C=C position-specific fragment ions from *N*-acyl chains were detected, as confirmed using UltimateSPLASH standards. Thus, the label "MSL+DB" was assigned to sphingolipids when the *N*-acyl chain C=C positions were determined."

7. *Figure 2c,d: There is no apparent reason why there are no MS2 spectra of CE, LPG and LPI available, since PG and PI are reported with the same measurement conditions. Please elaborate on this point. In addition, please clarify whether double bonds were assigned for the sphingoid base or only to the N-acyl chain for Cer and SM.*

Thank you for your comments. Upon reanalyzing the positive ion mode LC-MS/MS data for LPG and LPI, we found that the optimal ion forms for detection differed between the SCIEX ZenoTOF 7600 and Shimadzu LCMS-9030 systems. In *Nature Communications* **15**, 9903 (2024), where ZenoTOF 7600 was used, LPG and LPI were detected as protonated ions ($[M+H]^+$). However, under the Shimadzu LCMS-9030 conditions used in this study, LPG and LPI were detected as ammonium adducts ($[M+NH_4]^+$) rather than $[M+H]^+$. Since our OAD MS/MS annotation pipeline was initially constructed based on the SCIEX system, it did not account for the $[M+NH_4]^+$ adducts of LPG and LPI, leading to their absence in Figure 2c/d. In this revision, we have incorporated the $[M+NH_4]^+$ ion forms for LPG and LPI, allowing their successful characterization using OAcID MS/MS spectra (revised Figures 2c and 2d). However, cholesteryl esters (CE) were not characterized for a different reason. Severe in-source fragmentation under the Shimadzu LC-MS/MS conditions, rendering them undetectable. While lowering the interface temperature ($\sim 150^\circ\text{C}$) could mitigate this issue, it would also reduce the sensitivity of other lipid subclasses. Thus, CE results were excluded

from Figures 2c and 2d, and the revised figures do not include CE molecules. This clarification has been added to the manuscript (lines 222–225):

"Under the Shimadzu LC-MS/MS conditions employed in this study, significant in-source fragmentation prevented the detection of cholesteryl esters (CEs). Although lowering the interface temperature (~150°C) could potentially mitigate this issue, it would simultaneously compromise the sensitivity of other lipid subclasses. To maintain analytical robustness, CE data were therefore excluded from the evaluation."

For the assignment of C=C positions in sphingolipids, please refer to our response to minor comment #6.

8. Line 190: *'the annotation results of SM, PE, and TG molecules became species level (SL)' – this is inconsistent with Figure 2c, where all TG molecules are reported at least at MSL level.*

"TG" has been removed for consistency with Figure 2c.

9. *Optimization of fragmentation rules for Cer at lines 214-216: The EquiSPLASH mixture contains only one Cer compound (Cer 18:1;O2-d7/15:0). It is not apparent why the UltimateSPLASH ONE mixture was not used for this purpose. This standard mixture contains five different Cer molecules, which would give more confidence in the derived decision tree that was extrapolated to sulfatides.*

Your concerns have been addressed. The automated annotation of ceramides was optimized using both EquiSPLASH and UltimateSPLASH standard mixtures. Accordingly, we have revised the sentence as follows: "Additionally, the C=C positions of SHexCers on the sphingosine backbone were annotated based on the fragmentation rules of Cers optimized using standard mixtures of both EquiSPLASH and UltimateSPLASH." (lines 258–260).

10. *Lines 247-249 and 293-296: Did you intend to say: "OAcID-MS/MS is advantageous compared to EAD in a situation of two co-eluting C=C isomers, because..."? In addition, the claim that both coeluting isomers cannot be identified by EAD requires stronger evidence. The two spectra appear quite clearly distinguishable by the two highlighted fragments in the referenced Figure S6c, where it is shown that fragments produced by the McLafferty rearrangement allow for the assignment of the double bond positions. And from the intensity of these fragments, it is apparent that they can be distinguished. Please either tone down this claim or provide stronger evidence, e.g., by performing a dilution series between two such isomeric standards to determine the ratio at which the lower abundant isomer is still detectable with either method.*

As suggested, we have clarified our claim while appropriately adjusting its tone. EAD MS/MS allows the assignment of C=C positions in co-eluting isomers by recognizing diagnostic ions or V-shaped patterns. Our previous MS-DIAL 5 study indicated that EAD MS/MS could characterize a higher amount of C=C structural isomer in a mixture. However, in EAD MS/MS, the homolytic cleavage of all C–C bonds in acyl chains generates three fragment ion types (H-loss, radical, and H-gain), making C=C position assignments more reliant on quantitative intensity differences

(e.g., comparing H-loss and radical ion intensities relative to the V-shaped valley). This reliance on quantitative comparison can complicate distinguishing isomers, particularly when their abundance ratios vary. In contrast, OAD and OAciD MS/MS provide distinct diagnostic ions, as demonstrated for PC 16:0_16:1(Δ 9) and PC 16:0_16:1(Δ 7). Unlike EAD, where characteristic V-shaped patterns can be obscured in co-elution, OAciD produces non-overlapping diagnostic fragment ions, allowing for clearer annotation. This advantage extends to estimating the relative abundances of co-eluting isomers, where the standard deviations in diagnostic product-ion intensity ratios are larger in EAD MS/MS than in OAD MS/MS as described in *Nature Communications* **15**, 9903 (2024). To reflect this distinction more clearly, we have revised the sentence as follows (lines 291–300):

"OAciD MS/MS offers a distinct advantage over EAD when analyzing co-eluted C=C isomers. For instance, in the case of PC 16:0_16:1(Δ 7) and PC 16:0_16:1(Δ 9), EAD MS/MS annotation requires both diagnostic m/z values and the characteristic V-shaped pattern of product ion intensities. This pattern results from highly efficient fragment formation via homolytic cleavage at the allylic position (stabilized by delocalized electrons) and the McLafferty rearrangement (involving hydrogen transfer at the γ -position). Since electrons disrupt each carbon-carbon covalent bond in fatty acyl chains, producing H-loss, radical, and H-gain fragment ions, the characteristic V-shaped pattern is often obscured when isomers co-elute (Figure S6c). In contrast, OAciD produces distinct, non-overlapping diagnostic fragment ions, enabling more precise annotation of these isomers (Figures 4a and S7). This advantage is further highlighted in abundance ratio estimations of co-eluted isomers, as standard deviations in diagnostic product-ion intensity ratios are larger in EAD MS/MS than in OAD MS/MS⁴⁰."

11. Line 407: Please change "detailed position" to "detailed C=C position" or "detailed double bond position" to avoid confusion with *sn*-position assignment.

Thank you. We have updated the wording to "detailed C=C position".

12. Line 437: 'Circles represent the linear range of the MS1 peak with $R_2 > 0.9800$ (Table S1).'; Shouldn't this be Table S2?

Thank you for your attention to detail. The table reference has been corrected.

13. Suggestion for line 437: Please change "The closed circles" to "The fully filled circles ..."

Thank you. We have implemented this revision.

Reviewer #2 (Remarks to the Author):

I co-reviewed this manuscript with one of the reviewers who provided the listed reports. This is part of the Communications Chemistry initiative to facilitate training in peer review and to provide appropriate recognition for Early Career Researchers who co-review manuscripts.

We appreciate your clarification.

Reviewer #3 (Remarks to the Author):

This manuscript describes the use of combined collision induced dissociation (CID) and oxygen attachment dissociation (OAD) tandem mass spectrometry (MS/MS) methods to enable high confidence structural identification in lipidomics workflows. Building off prior work, this combined approach is novel and is also more readily applied to negative ion mode lipid analysis. A wide variety of lipid classes, fatty acyl tails, and double bond positions are identified. The work is nicely contextualized by analyzing complex extracts from marmoset brain. While sufficient replicates are not analyzed to draw biological conclusions comparing young and old brains, the lipid observations are nicely contextualized. The figures, though complex and a lot to digest, are informative and clear. There are a few instances where comparisons and contextualization to existing literature would be relevant (e.g., references in the Introduction, comparisons to MSI). Please see below for specific questions and comments:

We appreciate your positive comments and suggestions, which have significantly improved our manuscript. Our detailed responses are as follows.

1. Several references should probably be added:

a. OzID: The Blanksby lab pioneered OzID and has published 10-20 papers on this technique, but their work is not referenced. In particular, several recent reports use OzID in a similar manner to the workflow proposed in this manuscript: <https://doi.org/10.1002/anie.202316793> and <https://doi.org/10.1038/s41467-023-39617-9>

b. Ion/ion reactions: Gas-phase ion/ion reactions have also been used to reveal lipid sn-, double bond, and stereochemistry positions, and are worth mentioning (mainly from the McLuckey and Prentice labs):

i. <https://doi.org/10.1021/acs.analchem.0c02350>

ii. <https://doi.org/10.1002/anse.202300063>

iii. <https://doi.org/10.1021/acs.analchem.3c03804>

iv. <https://doi.org/10.1021/jasms.4c00368>

v. <https://doi.org/10.1021/acs.analchem.8b03441>

vi. <https://doi.org/10.1021/acs.analchem.9b04376>

vii. <https://doi.org/10.1021/ac400190k>

viii. <https://doi.org/10.1021/acs.analchem.5b02243>

c. EAD: The Prentice and Kane labs have also contributed to lipid EAD:

- i. <https://doi.org/10.1002/jms.3698>
- ii. <https://doi.org/10.1016/j.ijms.2020.116338>
- iii. <https://doi.org/10.1016/j.ijms.2022.116998>
- iv. <https://doi.org/10.1021/acs.analchem.3c03077>

Thank you for sharing our key references on OzID, ion/ion reactions, and EID. We have incorporated these into the Introduction section, providing brief summaries of each technique (lines 59–66 for gas-phase ion/ion reactions, 70–74 for EAD, and 78–82 for OzID). The added sentences are highlighted in yellow.

2. *Nomenclature: The hyphen between “CID-MS/MS” and “OAD-MS/MS” should be removed. Hyphens in these types of abbreviations are used to denote orthogonal analytical techniques coupled together (e.g., LC-MS/MS and GC-MS/MS). CID is not being coupled to MS/MS in these instances; it is the type of MS/MS being performed. “CID MS/MS” or “MS/MS via CID” would be more appropriate terminology.*

Thank you. We have adopted the suggested terminologies.

3. *Could the authors describe (and perhaps show in the Supplementary Information) the mechanism for OAD? This would facilitate understanding the fragmentation discussion.*

The fragmentation mechanism of OAD is described in the Discussion section and further addressed in response to comment #8 from Reviewer #4. (lines 359–373)

"The 96.13 Da loss occurs when OH \cdot adds to the Δ 10 position, initiating radical-induced carbon-carbon bond cleavage, which subsequently generates an enol group-containing fragment, whereas the 138.14 Da loss is initiated by O addition at the Δ 9 position, leading to cleavage between Δ 8 and Δ 9 positions (Figure 1a)³⁸. These major fragments were detected independently of lipid subclass and MS polarity and remained detectable even at high collision energies, provided that the charge remained on the fragment containing the fatty acyl chains with double bonds."

4. *The authors attribute the high fragmentation efficiency of sodium adducted ions to “the strong charge bias introduced by metal ions like sodium, compared to proton or ammonium ions.” What does the term “charge bias” mean and how does this affect the efficiency of the fragmentation mechanism?*

The term "charge bias" refers to the degree of charge localization at specific functional groups within lipid molecules. Protons exhibit minimal localization, binding flexibly to multiple nucleophilic sites such as carbonyl or ester oxygen atoms, leading to diffuse fragmentation. Ammonium ions show moderate localization due to partial delocalization, leading to broader fragmentation patterns. In contrast, sodium ions have a high charge density and preferentially bind to negatively charged sites like phosphate groups, promoting selective bond cleavage and producing distinct

fragmentation patterns. These explanations have been added to the Discussion section and highlighted in yellow. (lines 344–348)

5. *What is the nominal pressure in the collision cell, and what is the partial pressure of water? It seems likely that the H₂O versus Ar experiments are really comparing background N₂ versus Ar gas behavior (i.e., the ions are not largely undergoing CID via collisions with H₂O). The higher mass of Ar would also support a lower collision energy, as the energy partitioning into the ion would be more efficient than with a lighter neutral target like N₂.*

In the OAD setting, water vapor was introduced at 0.3 mL/min, resulting in a collision cell pressure of approximately 0.2 Pa. Based on the LCMS-9030 system specifications, the partial pressure of Ar was 0.04 Pa when the inlet pressure was set to 17 kPa (low setting). Additionally, hydrogen gas was introduced at 0.04 Pa to reduce oxidation by O/OH radicals on the quadrupole electrodes, preventing insulating layers and ensuring stable potential. Although the OAcID system does not allow selective fragmentation solely via water vapor, water remains the primary collision gas. In contrast, when argon was used, its partial pressure in the collision cell was 0.2 Pa with an inlet pressure of 230 kPa. Although we do not fully understand your concern, we believe nitrogen did not contribute to the observed fragmentation.

As you mentioned, CID fragmentation efficiency depends on the collision gas mass due to energy partitioning between the lipid ion and the gas molecule. In the laboratory frame, collision energy is determined by the acceleration voltage applied to the mass spectrometer. However, the energy available for fragmentation depends on the relative masses of the lipid ion and the collision gas. Heavier gases like argon (39.96 Da) transfer more energy to lipid ions upon collision, whereas lighter gases like water (18.01 Da) impart less energy, requiring a higher initial collision energy to achieve comparable fragmentation. Although the observed energy difference may align more closely with theoretical values if pure water vapor were used, our findings are consistent with the experimental result that lipid fragmentation with water vapor requires approximately 10 eV more collision energy than with argon. Higher collision energy compensates for reduced energy transfer efficiency, ensuring sufficient energy for bond cleavage.

These partial pressures have been clarified in the Methods section, and the above clarification has been added to the Discussion section. (lines 374–389 and 464–468)

6. *The statement that “little work has been done on the lipid profiling of mammalian brain regions” seems a bit odd (there are dozens of MALDI and DESI mass spectrometry imaging studies that have mapped lipid distributions in mammalian brains). How do the lipid abundances observed here compare to previous MSI studies?*

From the perspective of lipid coverage and quantitative reliability, LC-MS data provide higher precision than MSI data. However, MSI is valuable for profiling lipid distributions in anatomically inaccessible regions where direct sampling is not feasible. Our investigation suggests that RPLC-based untargeted lipidomics studies of the marmoset brain are relatively scarce compared to MSI-based studies. To clarify the advantage of our study, we have revised the

related sentences as follows (lines 251–252):

"However, little work has been done on reversed-phase LC (RPLC)/MS-based untargeted lipidomics of marmoset brain regions."

While MSI studies offer valuable insights, our study primarily compares lipid profiles with previously reported brain region lipidome data. Additionally, we analyzed gene expression patterns in marmoset brain regions to enhance the contextual relevance of our findings. In response to your suggestion, we have referenced the MALDI imaging studies cited in this article. These studies examine *sn*-position localization (e.g., PC 16:0/18:1 vs. PC 18:1/16:0) but do not address C=C position characterization due to MSI spectral complexity. Since OAcID does not resolve *sn*-positions, direct comparisons between MSI studies and our findings remain challenging. However, our approach provides complementary insights, particularly in double-bond characterization, which is less accessible through MSI.

7. It appears that positional sn- isomers are not resolved in this method (e.g., PC 16:0/18:1 versus PC 18:1/16:0), which should be mentioned.

We have added the following sentence to the Introduction for clarity (lines 106–108):

"Although OAD, unlike EAD, cannot determine *sn*-positions, CID-specific fragment ions related to polar head groups and FA compositions can still be obtained under OAD collision-cell conditions"

Reviewer #4 (Remarks to the Author):

The authors present results of combining the oxygen-addition dissociation (developed by some of them) with traditional collision-induced dissociation for lipid structural characterisation. The combined fragmentation patterns, notably in negative ion mode, are successfully demonstrated to yield a richer structural assignment than utilising either method in isolation; including the identification of carbon-carbon double bonds and discrimination between potential lipid isomers. The method is deployed against a sizeable set of lipid standards and biological extracts. The experiments are largely consistent with the data acquired and data analysis has been incorporated into MS-DIAL; a widely used software suite for lipid –and other metabolite– analysis. The manuscript should be published after the authors consider the following suggestions and corrections:

Thank you for your positive comments on our manuscript. Below are our point-by-point responses in blue font.

(1) The literature review and introduction are superficial. For example, the alternative and well-established methods of OzID and UVPD are mentioned but the original papers introducing these methods for lipid analysis are not cited. The papers cited for OzID are actually not comparable to the methods described herein as the ozonolysis is not conducted on mass selected ions. The UVPD citation references a combination method not the comparable UVPD of mass-selected ions. Both methods have been widely explored for positive and negative ions in a similar manner to much of the OAD discussion in this paper. The primary literature should be carefully reviewed and summarised in the context of the current study.

Thank you. We have revised descriptions of OzID, UVPD, and ion/ion reactions and summarized them in the Introduction (lines 59–66 for gas-phase ion/ion reactions, 78–82 for OzID, and 82–87 for UVPD). See also our response to Reviewer #3, comment #1.

(2) Similar to (1), the advantages of using CID with OzID (DOI: 10.1039/c3an01712e) and UVPD (DOI: 10.1021/jacs.7b06416) have been described before and the advantages of these combinations for full structural assignment are described in detail. In addition, a combination CID-OzID experiment (i.e., without MS3 mass-selection) has also been described (DOI: 10.1016/j.ijms.2018.05.016) that is highly relevant to the composite OAD-CID experiment outlined here. These prior studies find that composite dissociation can still resolve product ions arising from sequential CID/OzID processes, i.e., ozonolysis of CID product ions. It is unclear whether any such ions are observed here?

Although fragment ions related to polar head structures, fatty acyl chains, and C=C positions may appear similar across methods, differences in fragmentation mechanisms affect data analysis. In EAD, many product ions remain uninterpreted and are not used for annotation. The COzID technique (DOI: 10.1039/c3an01712e) combines CID and OzID to differentiate *sn*-isomers of alkali-adducted phospholipids by inducing ozonolysis at newly formed double bonds at the *sn*-2 position after CID-mediated head group loss. Although COzID and OAcID may appear similar,

their objectives and effects differ fundamentally.

The OzID and COzID combination (DOI: 10.1016/j.ijms.2018.05.016) optimized isomer and isobar separation using two injections under method-specific conditions, reducing throughput. In contrast, our approach offers a different advantage. The referenced study noted that "while combining CID and OzID is effective in obtaining near-complete structural identification of phospholipids, **it can further complicate spectra** and impair the ability to identify low-abundance lipids". The OAciD system generates fewer uninterpreted fragments, simplifying interpretation.

The HCD/UVPD analysis (DOI: 10.1021/jacs.7b06416) distinguished two C=C positional isomers like PC 18:1(*n*-12)/18:1(*n*-12) and PC 18:1(*n*-9)/18:1(*n*-9) in sodium ion form, but potential misannotations remain a concern due to non-specific fragments. (We cannot fully assess the specific pattern of misannotations until we conduct the actual data analysis.)

A key advantage of OAciD is that it does not require sodium solutions for determining polar heads, fatty acyl chains, and C=C positions, maintaining the same workflow as conventional CID-based lipidomics in both positive and negative ion modes. Additionally, OAD produces fewer double bond-specific fragments than other C=C resolving techniques, reducing spectral complexity. This advantage led to the analysis of potential isomers in PC 16:0_16:1 (Figure 4a). However, OAciD cannot resolve *sn*-positions, requiring complementary techniques. This limitation should be considered alongside the method's strengths and study objectives.

These discussions have been incorporated into the Introduction and Discussion sections and are highlighted in yellow. (lines 78–95 and 416–418)

(3) Pg 2, line 1. *What are the energy storage materials?*

The term has been updated to "stored energy sources."

(4) Pg 2, line 45. *What is meant by "description in the quality and quantity of lipids"? this is unclear.*

The sentence has been revised as follows (lines 45–46):

"Dysregulation of lipid species abundance disrupts cellular homeostasis and may contribute to disease pathogenesis in animals."

(5) Pg 2, line 60. *The introduction of other techniques is confused as it seems to intertwine methods relying on external derivatization of lipids prior to mass spectrometry with those that use selective dissociation of underivatized lipids, namely, EAD, OzID, UVPD and OAD. This should be clarified and appropriately referenced. Similarly, the use of the term "specialised equipment" is specious here. All these techniques require specialised equipment. With the exception of OzID all have been commercialised. Commercialisation of a technology does not make it any less*

specialised.

This comment aligns with Major Comment #4 from Reviewer #1. To avoid confusion, we have removed the phrase "or specialized equipment."

(6) Pg 3, Line 89. The statement that negative ions are "a domain that has not been thoroughly explored for lipids" is incorrect. See any text book on lipid structural analysis (e.g., Murphy's Tandem Mass Spectrometry of Lipids) which are full of negative ion analysis.

Thank you for your comment. Reviewer #1 raised a similar concern (Minor Comment #3). We have revised the statements to: "To extend lipid coverage, we evaluated the fragmentation patterns and efficiency of OAD MS/MS in negative ion mode, marking its first application in OAD-based lipidomics for this polarity." (lines 119–121)

(7) Pg 3, line 113. What is "charge bias"?

Reviewer #3 raised a similar concern (Comment #4). "Charge bias" refers to the effects of charge localization on lipid fragmentation.

(8) Pg 9, line 305. The chemical descriptions of the radical chemistries are inaccurate. Hydroxyl radicals can add to a carbon-carbon bond with subsequent hydrogen atom abstraction (by oxygen or other scavengers) or direct elimination of hydrogen yields the new double bond.

Thank you for your comments. We have confirmed the possible fragmentation mechanism of OAD proposed by density functional theory calculations (H. Takahashi et al., *Anal. Chem.* **90**, 7230–7238 (2018)). Accordingly, we have revised the descriptions in the Discussion section, highlighting the changes in yellow (lines 368–373).

(9) Pg 9 (and elsewhere). One of the limitations of negative ion CID fragmentation of glycerolipids has been the inability to distinguish between BMP and PG isomers. It would be interesting to see whether the competitive CID-OAD spectra provide unique assignments. How are the authors sure they are assigning one and not the other?

Similar to conventional CID, OAciD did not generate unique fragment ions for BMP and PG in negative ion mode. We still relied on positive ion mode spectra to distinguish these structural isomers. The CID reactions with H₂O followed a similar trend to those with Ar, though higher collision energy was required for H₂O.